# Intrinsically fluorescent polyureas toward conformation-assisted metamorphosis, discoloration and intracellular drug delivery

Yeqiang Zhou[1,2], Fan Fan[1,2], Jinling Zhao[1], Zhaoding Wang[1], Rui Wang[1], Yi Zheng[1], Hang Liu[1], Chuan Peng[1], Jianshu Li [1], Hong Tan[1], Qiang Fu [1] & Mingming Ding [1] ✉

Peptidomimetic polymers have attracted increasing interest because of the advantages of facile synthesis, high molecular tunability, resistance to degradation, and low immunogenicity. However, the presence of non-native linkages compromises their ability to form higher ordered structures and protein-inspired functions. Here we report a class of amino acid-constructed polyureas with molecular weight- and solvent-dependent helical and sheet-like conformations as well as green fluorescent protein-mimic autofluorescence with aggregation-induced emission characteristics. The copolymers self-assemble into vesicles and nanotubes and exhibit H-bonding-mediated metamorphosis and discoloration behaviors. We show that these polymeric vehicles with ultrahigh stability, superfast responsivity and conformation-assisted cell internalization efficiency could act as an "on-off" switchable nanocarrier for specific intracellular drug delivery and effective cancer theranosis in vitro and in vivo. This work provides insights into the folding and hierarchical assembly of biomacromolecules, and a new generation of bioresponsive polymers and nonconventional luminescent aliphatic materials for diverse applications.

Natural amino acid-derived polymers are recognized as promising materials to understand the structures and functions of proteins and develop bioinspired functional biomaterials for use in biomimicry, drug delivery, and tissue engineering applications[1–3]. Among them, poly(amino acid)s (PAAs) have attracted increasing attention due to their good biocompatibility, nontoxic metabolized products, and specific secondary conformations such as α-helix and β-sheet[4–10]. Alternatively, a variety of synthetic peptidomimetic polymers (poly-β-peptide, polypeptoids, etc.) have also been developed via amino acid extension, polypeptide side chain substitution, or backbone modification[11–15]. The peptidomimetics hold the additional advantages of highly tunable molecular architecture, enhanced solubility, resistance to degradation, and low immunogenicity compared with their peptide counterparts. However, the incorporation of non-native

moieties may compromise their propensity to form higher-order hierarchical structures and protein-mimic biological functions.

Hydrogen-bonding (H-bonding) interaction is known as an important governing force in biology that dictates the secondary structures and folding specificity of proteins[16]. For example, metamorphic proteins undergo large-scale structural changes involving complete rearrangement of H-bonding networks and alterations in secondary structures[17]. Motivated by this rationale, the control of side-chain H-bonding ligands has been utilized to modulate the conformation of synthetic polypeptides[18]. As an interesting building block with strong H-bonding capacity, urea moiety containing one H-bonding acceptor (C=O group) and two donors (N–H groups) is capable of forming single and bidentate intermolecular H-bonding interactions[19,20]. In fact, sequence-defined oligoureas have been

---

[1]College of Polymer Science and Engineering, State Key Laboratory of Polymer Materials Engineering, Sichuan University, Chengdu 610065, China. [2]These authors contributed equally: Yeqiang Zhou, Fan Fan. ✉e-mail: dmmshx@scu.edu.cn

established as attractive peptidomimetic foldamers that adopt robust helical conformation stabilized by three-center H-bonds[21–23]. As a class of urea-rich polymers, polyureas (PUs) have been widely used as elastomers, coatings, adhesives, and biomaterials[24–26]. Nonetheless, PU-based functional peptidomimetics with ordered secondary conformations and conformation-mediated functionalities have been rarely reported.

Herein, we report a family of intrinsically fluorescent PUs constructed from natural amino acid derivatives. The polymers show molecular weight- and solvent-dependent conformations and aggregation-induced emission (AIE) properties. Moreover, the polymers self-assemble into vesicles and nanotube-like structures, which exhibit H-bonding-mediated size expanding and shrinking, metamorphosis, and discoloration behaviors. In addition, the polymeric vehicles show ultrahigh stability and superfast responsivity that enable an "on-off" switchable nanocarriers for potential drug delivery and theranosis applications.

## Results

### Molecular design and characteristics

The PUs were synthesized via a facile one-pot reaction using L-lysine ethyl ester diisocyanate (LDI), L-cystine dimethyl ester dihydrochloride (L-Cys·OMe·2HCl), and L-lysine ethyl ester dihydrochloride (L-Lys·OEt·2HCl) as monomers and polyethylene glycol monomethyl ether (mPEG5000) as an end-caping agent (Fig. 1a, Supplementary Figs. 1, 2). By changing the feed ratios of monomers, the length of hydrophobic chains could be tuned. The structures of the polymers were analyzed with Fourier transform infrared (FTIR) and nuclear magnetic resonance ($^1$H NMR and $^{13}$C NMR) (Supplementary Figs. 3–6) spectroscopy. The presence of urea C=O stretching vibration band (1600–1700 cm$^{-1}$) and the disappearance of isocyanate signal (2270 cm$^{-1}$) confirmed the successful synthesis of PUs (Supplementary Fig. 7). In the $^1$H NMR spectra (Supplementary Fig. 3), the characteristic peaks assigned to mPEG (3.50 ppm), Lys·OEt (4.08 ppm), and Cys·OMe residues (3.66 ppm) were observed. By integration of the peaks (Supplementary Table 1), the molecular weights and compositions of the polymers were calculated and listed in Table 1. All the polymers showed narrow molecular weight distributions (Đ 1.02–1.20) with single elution peaks in the gel permeation chromatography (GPC) traces (Supplementary Fig. 8). To verify the triblock architecture of PUs, we synthesized a control polymer containing diblock and triblock mixtures (DTPU) by feeding half amount of mPEG (Supplementary Fig. 9). As expected, DTPU exhibited much larger polydispersity (Đ 1.92) and bimodal molecular weight distribution (Supplementary Fig. 10). It is believed that diblock or non-PEGylated PUs possess one or two amine groups on the chain end due to the reaction of isocyanate-terminated prepolymers with water[27]. Therefore, we carried out an end-group analysis taking P$_3$ and DTPU as examples. The polymers were reacted with fluorescein isothiocyante (FITC) and subjected to UV–vis and fluorescence measurements. Apparently, DTPU displayed distinct UV absorption at 490 nm and a fluorescence emission peak at 552 nm (Supplementary Fig. 11), suggesting that it has been successfully labeled with FITC. In contrast, P$_3$ did not show noticeable FITC signal (Supplementary Fig. 11), thus confirming the PEGylated triblock structure of PUs that did not have any reactive groups.

### Self-assembly of PUs

Because of their amphiphilic character, the PUs can self-assemble into nanoparticles in an aqueous solution (Fig. 1a). Counterintuitively, the size of the assemblies decreased from 119 to 60 nm with increasing length of hydrophobic segments (Table 1). Moreover, as the hydrophobic chain length increased to 81 (P$_4$), a bimodal size distribution was observed (Fig. 1b). To understand the size change, the assemblies were observed by transmission electron microscope (TEM). The result indicates that PUs with chain lengths of 21–41 (P$_1$–P$_3$) self-assembled into vesicles, while P$_4$ formed non-spherical structures with diameters and lengths in the range of 10–30 and 93–132 nm, respectively (Fig. 1c, Supplementary Figs. 12 and 13). Further cryo-scanning electron microscopy (cryo-SEM) image of P$_4$ presents a number of open-ended tube-like particles with large aspect ratios (Supplementary Fig. 14), suggesting a possible hollow interior structure. Small-angle X-ray scattering (SAXS) provided more information on the morphology of PU assemblies. The scattering pattern of P$_2$ presents a gradient of approximately −2 at low q and corresponds to a classic model of vesicle (Supplementary Fig. 15a)[9], while the SAXS data of P$_4$ shows regular oscillations on the decay, which could be reasonably fitted to a hollow cylinder model (Supplementary Fig. 15b)[14]. The formation of nanotube-like structures warrants further investigation. The assembled morphology was also determined by a fluorescence technique using rhodamine 6G (R6G) and doxorubicin hydrochloride (DOX·HCl) as hydrophilic probes. With the incorporation of R6G in the PU dispersions, the fluorescence intensity of the dye was much lower than that of free R6G dissolved in water (Fig. 1d), demonstrating a self-quenching effect resulted from the high local concentration of dyes within the vesicular or tubular interior[10]. Moreover, the DOX fluorescence was detected in both emission spectra (Supplementary Fig. 16) and confocal laser scanning microscopy (CLSM) images with typical spherical and tube-like architectures (Fig. 1c), which agrees well with TEM observation. These results reveal that the polymeric assemblies are capable of providing hydrophilic pockets for loading water soluble agents[28,29]. However, the nanovessels cannot accommodate hydrophobic dyes as indicated by a pyrene encapsulation experiment (Supplementary Fig. 17), possibly due to the presence of tightly packed H-bonding in the membrane that did not allow guest insertion[30].

The unusual size change of PU assemblies seems contradictory to traditional theory since the increase of hydrophobic chain length may reduce the interfacial curvature and result in the formation of larger aggregates[31]. Keeping in view the strong H-bonding capacity of PUs, we envision that the enhanced H-bond may compact the vesicular membranes and shrink the particles. To confirm this hypothesis, the assemblies were treated with different amounts of trifluoroacetic acid (TFA) that destabilized the H-bonding interactions[32]. Note that the TFA treatment did not the disrupt the molecular integrity of PUs (Supplementary Fig. 18). As seen in Fig. 1e, f, all the assemblies exhibited a dramatic size increase by 5–30 folds after TFA treatment. In particular, P$_1$, P$_2$, and P$_3$ showed critical TFA concentrations for size change of 4, 6, and 10%, respectively, suggesting that the copolymers with longer PU segments have stronger H-bond that needs more TFA to disturb (Supplementary Fig. 19). TEM and CLSM imaging confirmed that all the assemblies maintained their vesicular structures in the presence of TFA, where the swollen membrane structure could be readily labeled with a hydrophobic probe FITC due to the breakage of dense H-bonds (Fig. 1g). In particular, three-dimensional (3D) reconstruction of CLSM images allowed identification of vesicular membrane structures from the colocalization of DOX and FITC fluorescence (Supplementary Fig. 20)[33]. As expected, when enough TFA was added to completely break the H-bonds, the vesicular size increased with an increase of hydrophobic chain length (Fig. 1e, f), which is in agreement with traditional self-assembly theory. The results verify that H-bonding interaction played a dominant role in the unusual size changes of PU assemblies. Interestingly, the particle sizes shrunk to original diameters after removal of TFA by dialysis, and this process could be repeated many times (Fig. 1h, Supplementary Fig. 21). The H-bonding-assisted vesicular respiring phenomenon in a size range up to 30 folds has been rarely reported in the literature[34], which holds great promise in biosensing, nanoreactor and diagnosis applications.

### Molecular conformations and metamorphosis

On the other hand, considering that P$_4$ displayed an abnormal morphology compared with other PUs, we wonder whether these

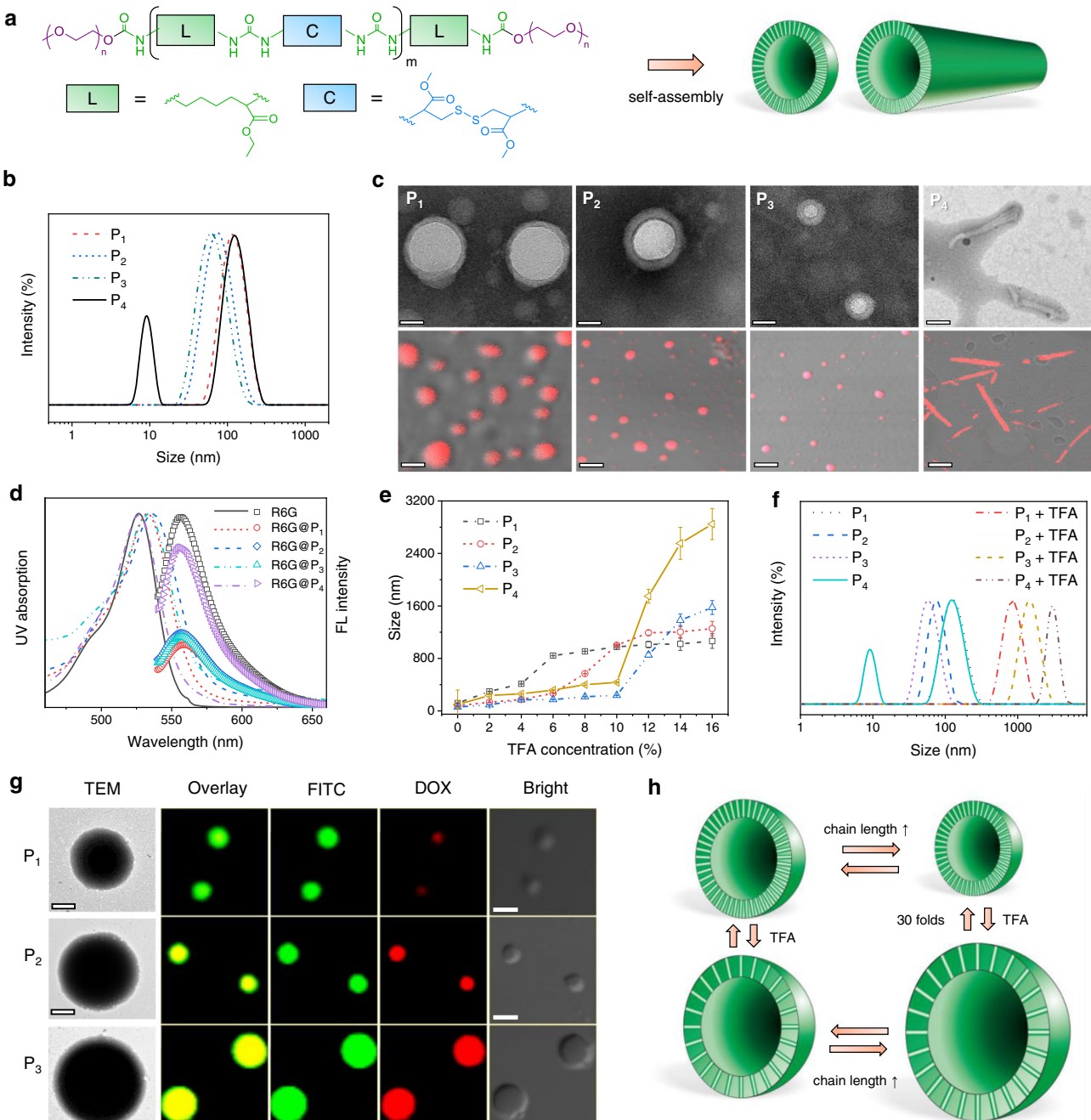

**Fig. 1 | Self-assembly of PUs. a** Schematic illustration of PUs and their assemblies. **b** Size distribution profiles of PUs determined by dynamic light scattering (DLS). **c** TEM (upper) and CLSM (lower) images of PU assemblies. The scale bars in TEM images are 50 nm and full TEM images are shown in Supplementary Fig. 10. For CLSM, the assemblies were encapsulated with DOX·HCl (red). The scale bars are 100 nm. Experiments were repeated three times independently with similar results. **d** UV–vis spectra (left) and fluorescence emission spectra (right, $\lambda_{ex}$ = 526 nm) of R6G dissolved in water and those encapsulated in PU assemblies (R6G@PUs) at the same concentration. **e** Size changes of PU assemblies treated with different concentrations of TFA. Data are presented as the mean ± standard deviation (SD) ($n$ = 3 independent measurements). **f** Size distributions of PU assemblies before and after TFA treatment (16 $v$%). Experiments were repeated three times independently with similar results. **g** TEM (left) and CLSM (right) images of PU assemblies treated with TFA. For CLSM, the assemblies were loaded with DOX·HCl (red) and FITC (green). The scale bars are 0.5 and 2 μm for TEM and CLSM, respectively. Experiments were repeated three times independently with similar results. **h** Schematic illustration of PU vesicles with or without TFA treatment.

polymers possess different secondary conformations, as chiral tyrosine-constructed polyurethanes have shown helical configurations[35]. With this in mind, we analyzed the polymeric assemblies with circular dichroism (CD). Of interest, P₁, P₂, and P₃ presented a negative peak at -196 nm followed by a positive band around 215 nm, which is in accordance with random coil structures. In contrast, P₄ displayed a CD pattern similar to that of β-sheet conformation, with a positive band at 200 nm and a single minimum at 218 nm (Fig. 2a)[10,36,37]. For comparison, we synthesized racemic PUs (DL-P₁ and DL-P₄) using LDI and D-Cys·OMe (Table 1, Supplementary Figs. 1, 22, and 23). CD characterization demonstrated that both samples adopted random coil structures in water (Supplementary Fig. 24). The results suggest that the conformation was influenced by the chirality and the number of segments. To further explore the unexpected conformation, we performed a thioflavin T (ThT) binding assay. ThT displays a fluorescence response when bound to sheet-rich

**Table 1 | Characteristics of polyureas and their assemblies**

| Sample[a] | Block numbers[b] | C[b] | L[b] | $M_{n1}$[b] | $M_{n2}$[c] | $M_w$[c] | Đ[c] | Size (nm)[d] | ZP (mV)[d] | PDI[d] |
|---|---|---|---|---|---|---|---|---|---|---|
| $P_1$ | 21 | 10 | 11 | 15,504 | 12,265 | 12,541 | 1.02 | 119 | −25.2 | 0.20 |
| $P_2$ | 31 | 15 | 16 | 17,688 | 12,829 | 13,900 | 1.08 | 80 | −33.5 | 0.17 |
| $P_3$ | 41 | 20 | 21 | 20,664 | 12,128 | 13,405 | 1.11 | 60 | −22.3 | 0.19 |
| $P_4$ | 81 | 40 | 41 | 29,840 | 16,611 | 20,010 | 1.20 | 97 | −0.46 | 0.35 |
| $P_5$ | 31 | 0 | 31 | 16,200 | 11,975 | 12,260 | 1.02 | 125 | −19.4 | 0.08 |
| DL-$P_1$ | 19 | 9 | 10 | 14,027 | 10,575 | 14,993 | 1.42 | 82 | −22.7 | 0.11 |
| DL-$P_4$ | 79 | 39 | 40 | 28,384 | 9222 | 15,226 | 1.65 | 112 | −21.3 | 0.03 |

[a]$P_1$, $P_2$, $P_3$, $P_4$, and $P_5$ represent PUs with different block numbers. DL-$P_1$ and DL-$P_4$ indicate racemic PUs.
[b]The block numbers, Cys·OMe residues (C), Lys·OEt residues (L), and molecular weights calculated by integration of NMR peaks.
[c]Molecular weights and molecular weight distributions (Đ) measured by GPC. Đ was calculated by the ratio between the weight average and the number of average molecular weights ($M_w/M_{n2}$).
[d]Size, zeta potential (ZP), and polydispersity (PDI) determined using a Zetasizer Nano ZS instrument at an angle of 90°.

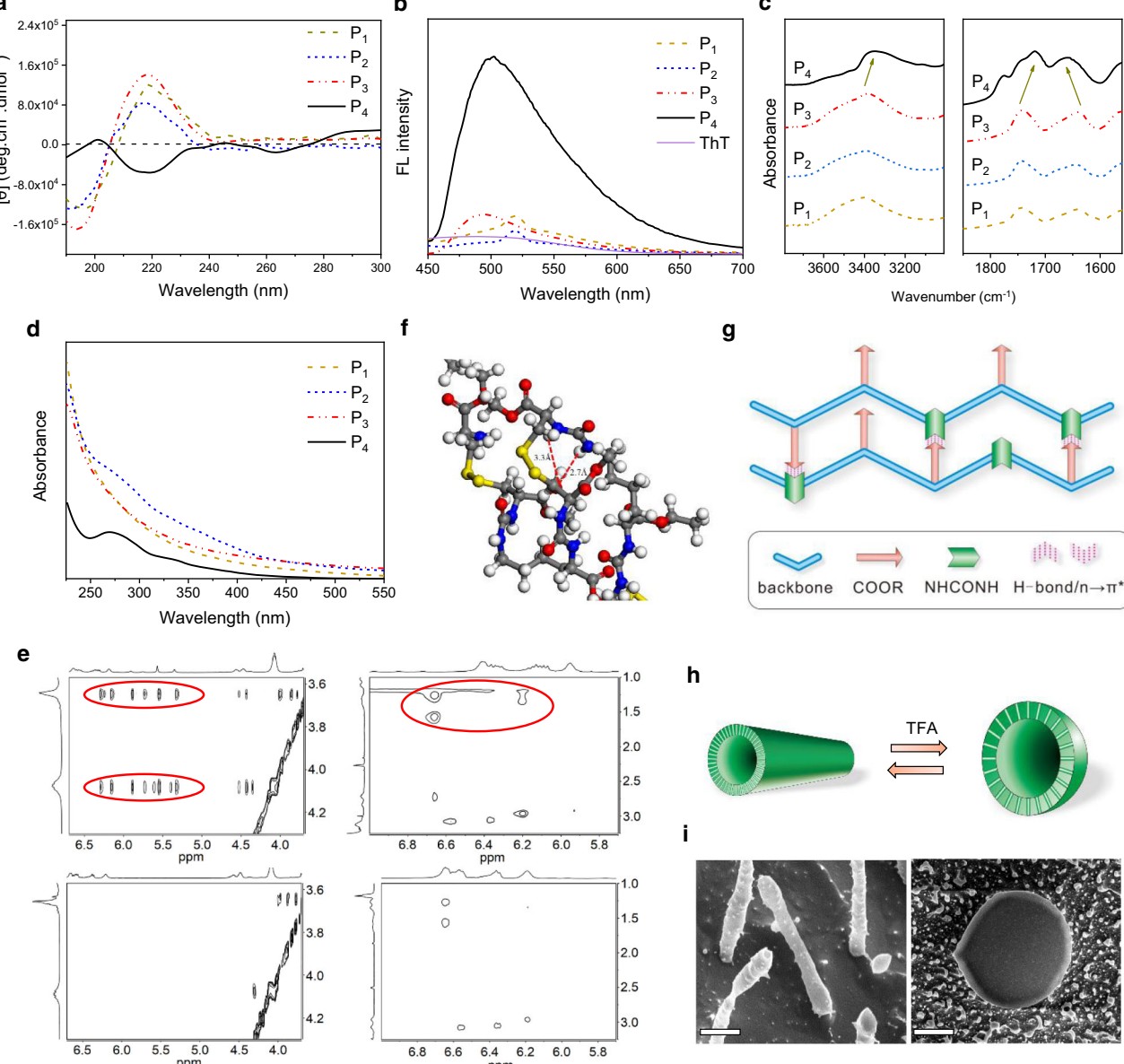

**Fig. 2 | Conformation-assisted metamorphosis. a** CD spectra of PU assemblies. **b** Fluorescence emission spectra ($\lambda_{ex} = 440$ nm) of PU assemblies incubated with 100 μM of ThT. **c** FTIR spectra of PUs in the NH (left) and C=O (right) stretching regions. **d** UV–vis spectra of PU assemblies. **e** $^1$H–$^1$H NOESY spectra of $P_4$ (upper panel) and $P_3$ (lower panel). **f** Representative molecular dynamics simulation images based on energy minimization for $P_4$. **g** Schematic illustration of molecular conformation of $P_4$. **h** Schematic illustration of reversible nanotube-to-vesicle transitions of $P_4$ with TFA treatments. **i** Cryo-SEM images of $P_4$ assemblies before and after TFA treatment (16 𝑣%). The left and right scale bars are 100 nm and 1 μm, respectively. Experiments were repeated three times independently with similar results.

structures and has been established as a standard tool for detecting β-amyloid aggregates in solution[38,39]. As shown in Fig. 2b, ThT exhibited a strikingly increased emission at 485 nm in the presence of $P_4$, and that incubated with other polymers showed negligible fluorescence enhancement. The result confirms the formation of sheet-like architectures by $P_4$. To understand the mechanism of the conformation, the H-bonding of the polymers was analyzed by FTIR. As seen in Fig. 2c, The NH stretching vibration of $P_1$, $P_2$, $P_3$ was located at 3400 cm$^{-1}$, and that of $P_4$ was downshifted to 3345 cm$^{-1}$, suggesting a strengthened H-bonding interaction in $P_4$[40]. Interestingly, a red-shifted ester carbonyl vibrational band (from 1741 to 1719 cm$^{-1}$) and a blue-shifted urea carbonyl peak (from 1637 to 1659 cm$^{-1}$) were observed for $P_4$ (Fig. 2c). The peaks were divided by Gauss formula and the contents of H-bonded ester/urea carbonyl groups were estimated to be 9/51% and 41/27%, respectively, for $P_3$ and $P_4$ (Supplementary Fig. 25 and Supplementary Table 2). The results indicate the presence of strong H-bonding interactions among the urea moieties in $P_1$–$P_3$, whereas those occurred mainly between urea linkages and ester carbonyl groups in the side chains of $P_4$. This is the reason for the formation of different secondary conformations. Moreover, the close contact and electron delocalization of the carbonyl moieties may also allow for a carbonyl-carbonyl n→π* interaction[41–43], as evidenced by a characteristic UV–Vis absorption of $P_4$ at 270 nm (Fig. 2d). Further $^1$H–$^1$H nuclear Overhauser enhancement spectroscopy (NOESY) test showed distinct cross-peaks between urea protons (5.5–6.3 ppm) and methyl and ethyl ester groups (3.66 and 4.08 ppm) in the side chains of $P_4$, and these NOE signals were negligible in the spectra of $P_3$ (Fig. 2e). The result implies that the urea linkages and ester moieties were in a closer proximity in $P_4$ than those in $P_3$, thus verifying the formation of H-bonding and n→π* interactions between these groups. To further assess the conformation propensity of PUs, we conducted molecular dynamics (MD) simulations on $P_2$ and $P_4$ constructs. It was predicted that $P_4$ preferred a sheet-like conformation, where the H-bond formed between urea linkages and carbonyl groups of side ester groups with distances of 2.7–3.3 Å (Fig. 2f, g)[44]. In contrast, $P_2$ showed a random coil structure (Supplementary Fig. 26). The results are in good agreement with CD, ThT, and FTIR analyses. Noting that the molecular weight dependence of secondary conformations has also been widely reported in poly(amino acid)s, where a critical chain length is needed to form stable conformations[45–47]. In addition, it is known that in biology the β-sheet conformation of peptides may result in the aggregation of fiber-like proteins and lead to a variety of neurodegenerative diseases[48,49]. Herein, the formation of sheet-like conformation in PUs might be the cause of nanotube formation. To confirm this hypothesis, $P_4$ assembly was treated with TFA to destabilize the H-bond and conformation. As expected, a transition from nanotube-like structures to vesicles was observed by DLS, cryo-SEM, TEM, and CLSM experiments (Fig. 2i-g, Supplementary Fig. 27). Interestingly, the tube-to-vesicle transformation could also be found in the presence of structure-promoting solvents such as trifluoroethanol (TFE) and methanol (Supplementary Fig. 28)[50,51]. CD analysis confirmed that $P_4$ assemblies underwent a transition from sheet-like to helical structure after TFE or methanol addition, as evidenced by the appearance of two minima at 207 and 223 nm in the CD spectra (Supplementary Fig. 28). This may be because these solvents strengthened the intramolecular interactions and promoted the formation of helical conformations[50,52]. The exact mechanism requires further examination. Such a conformation-driven shape-shifting phenomenon in peptidomimetics is reminiscent of protein metamorphosis[53], which is helpful to understand the folding behaviors of biological macromolecules.

## Nonconventional fluorescence properties

Nonaromatic polyurethanes, polypeptides, and other aliphatic compounds have been shown to generate noticeable intrinsic fluorescence under suitable conditions based on a clustering-triggered emission (CTE) mechanism[54–59]. However, most of these polymers were fluorescent when concentrated or aggregated as powders and films but nonemissive in dilute solutions[56,60]. In this work, the compact H-bonding of PUs resulted in the close proximity of carbonyl moieties to form a "heterodox cluster", which gave rise to emission under UV irradiation (Fig. 3a)[61]. Of interest, the dilute PU solutions, assemblies, and films emitted cyan blue and green fluorescence in the presence of UV irradiation (Fig. 3b and Supplementary Fig. 29). The intrinsic fluorescence enabled direct observation of PU assemblies using CLSM. As seen in Supplementary Fig. 30, the fluorescent images present well-dispersed spherical and tube-like particles with hollow structures, which are consistent with TEM results (Fig. 1c). The PUs showed high fluorescence quantum yields up to 29% (Supplementary Fig. 31). By comparison, the monomers and their mixture were nonfluorescent, and racemic DL-PUs showed much weaker fluorescence and lower quantum yields than corresponding PUs (Supplementary Fig. 32), indicating that both the formation of polyurea structures and chirality contributed to the interesting fluorescent property. The maximal excitation (365 nm) and emissions (410–475 nm) of PUs were quite similar to those of well-known blue fluorescence proteins (BFP) ($\lambda_{ex}$ 383 nm, $\lambda_{em}$ 448 nm)[62]. Moreover, the fluorescence of PUs exhibited typical AIE characteristics, with the fluorescence intensity increasing gradually as poor solvents were added (Fig. 3c). It should be noted that Bonifacio et al. have observed pH-dependent weak fluorescence in polyurea dendrimers, and the light emission originated from the protonation of primary and tertiary amino groups[63]. In our work, the nontypical chromophores were associated with the amino acid-based and urea-collected carbonyl moieties (Fig. 3a).

More interestingly, the fluorescence of PUs was found conformation-dependent. With the transformation from random coil to sheet-like structures, a blue shift and enhancement of fluorescence emission was observed (Fig. 3b). This may be due to the fact that the formation of sheet-like structure increased the intermolecular H-bonding and chain rigidity of polymers and suppressed the non-radiation relaxation[61]. To confirm this point, $P_4$ assemblies were treated with TFA. While the conformation changed from sheet to random coil structure (Supplementary Fig. 33), the fluorescence of $P_4$ was found weakened and shifted to higher wavelengths in the presence of denaturant (Fig. 3d), and could be further reversed after removal of TFA (Fig. 3e). Such an H-bonding-mediated conformation transition accompanied by change in color and morphologies resembles jellyfish that exhibits switchable fluorescence behavior with swelling and shrinkage of membrane during breathing process[34]. In nature, the blue light emitting proteins purified from jellyfish can transfer its energy to green fluorescence proteins (GFP) and result in green emissions[62,64]. Herein, the protein-mimic PUs can also act as a donor for fluorescence resonance energy transfer (FRET) (Fig. 3f). As the polymeric vesicles encapsulated quinacrine dihydrochloride (QD) as a receptor, the idiopathic fluorescence of polymers could be transferred to QD and the color of the solutions turned from blue to green (Fig. 3g, h). To our knowledge, this is the first example of FRET process based on the nontypical auto-fluorescence of linear aliphatic polymers.

## Ultrahigh stability and superfast responsivity

Owing to the dense H-bonding interactions, the PU assemblies were highly stable. The diameters and size distributions were almost unchanged even when diluted more than 200 times (Fig. 4a and Supplementary Fig. 34). Moreover, they also showed much higher tolerance to methanol and sodium dodecyl sulfate (SDS) as compared with methoxy poly(ethylene glycol)-poly(ε-caprolactone) (mPEG-PCL) micelles that were quickly destructed under these harsh environments (Supplementary Fig. 35). On the other hand, the high content of disulfide linkages within the polymeric backbone enabled a high responsivity to intracellular level of GSH (10 mM), resulting in superfast and nearly complete release of hydrophilic payload R6G within

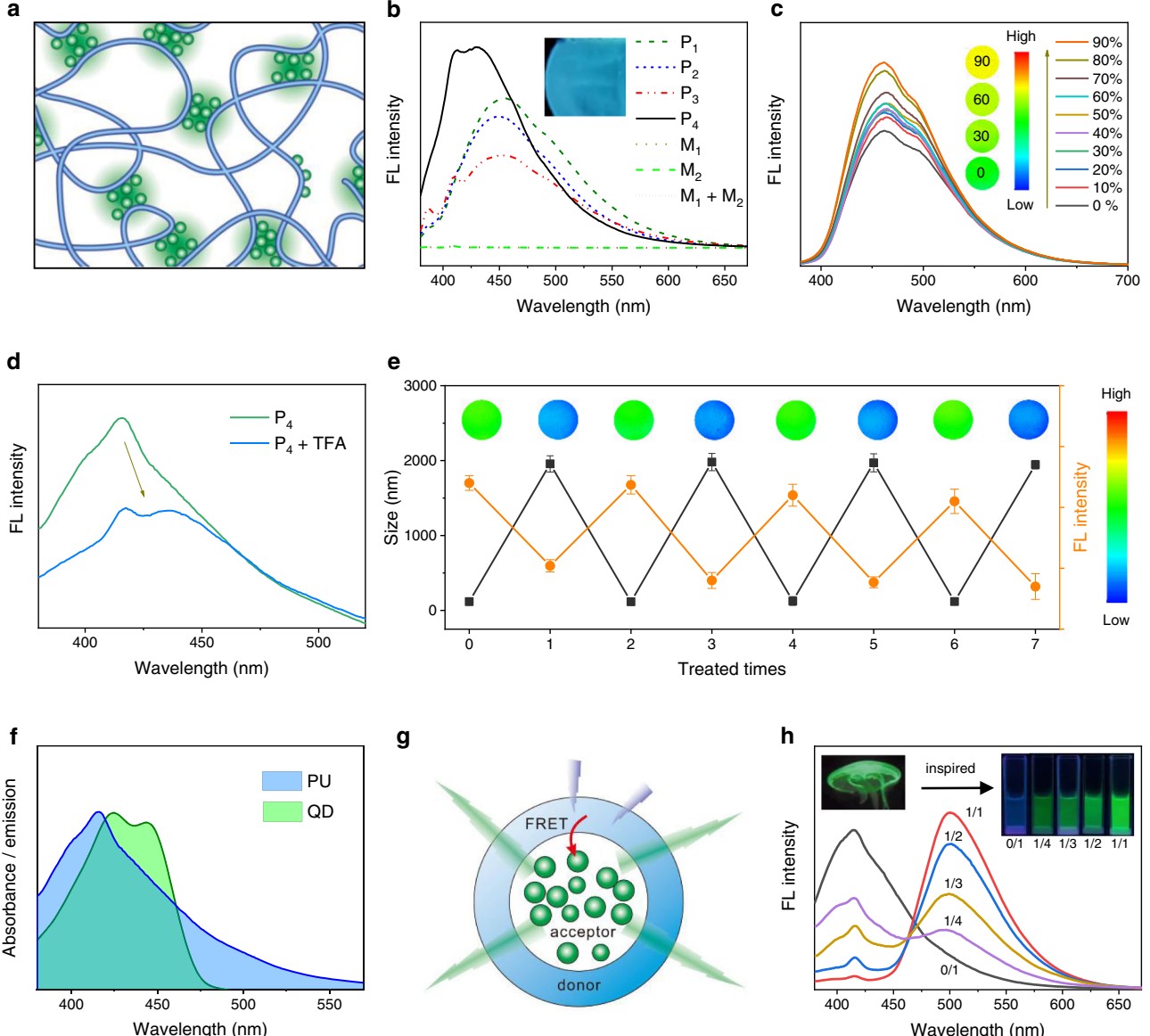

**Fig. 3 | Nonconventional fluorescence properties. a** Schematic illustration of the fluorescent chromophore of PUs. The green spherical clusters indicate the collection of carbonyl moieties. **b** Fluorescence spectra of PU assembles and monomers (1 mg mL⁻¹). $M_1$, $M_2$, and $M_1 + M_2$ represent Cys·OMe, L-Lys·OEt, and their mixture (molar ratio 1:1). The inset shows a photograph of PU film under a 365 nm UV lamp. **c** Fluorescence spectra of PU in DMF/CHCl$_3$ mixtures with CHCl$_3$ fractions from 0 to 90%. Insets show the emission images of PU in the mixtures with different CHCl$_3$ fractions (0, 30, 60, and 90%). **d** fluorescence spectra of $P_4$ assemblies before and after addition with 16% TFA. **e** Particle sizes and fluorescence intensity of PU assemblies before and after TFA treatments. TFA was added and removed repeatedly. Insets show corresponding fluorescent images of PU assemblies with or without TFA treatments. Data are presented as the mean ± SD ($n = 3$ independent measurements). **f** Spectral overlap between the emission spectrum of the donor (PU) and the absorbance spectrum of the acceptor (QD). **g** Schematic illustration of the FRET effect between the donor (PU assemblies) and the encapsulated acceptor (QD). **h** Fluorescence spectra of QD-loaded PU assemblies with different dye feeding ratios (QD/PU). Left inset shows a photograph of jellyfish, and right insets indicate the fluorescent images of assemblies under 365 nm UV light illumination.

10 min (Fig. 4b). The release rate was much faster than those reported for other redox-responsive drug delivery systems[1,65]. In contrast, no evident drug leakage was detected for $P_3$ vesicles incubated in a GSH-free medium and for control PU sample without disulfide bonds ($P_5$) in the presence of GSH (Fig. 4c, d, Supplementary Fig. 36). In addition, the FRET property between PU assemblies and their payloads can also serve as an effective and sensitive approach for probing the drug release profiles (Fig. 4e)[66,67]. We found that the emission of acceptor (QD) was quickly diminished within 10 min after GSH treatment (10 mM) (Fig. 4f), while that kept unchanged under normal conditions (Supplementary Fig. 37), further confirming the ultrahigh stability and hypersensitivity of PUs.

## Intracellular uptake and drug delivery

Next, we explored the potential of PU-based peptidomimetics in biomedical applications. By virtue of the intrinsic AIE fluorescence, the cell uptake and intracellular drug release of PU nanoparticles could be conveniently tracked in a label-free manner. We found that $P_4$ showed the greatest cell internalization efficiency, with relative fluorescence intensity in tumor cells two times higher than other polymers (Fig. 5a–c, Supplementary Fig. 38), possibly due to its sheet-like rigid conformation and nano-tubular morphology[68,69]. Moreover, the ultra-high responsivity of PU nanocarriers enabled a fast release of payloads within tumor cells and efficient delivery of DOX into cell nuclei (Fig. 5d–g)[29]. Particularly, the percentages of DOX fluorescence in cell

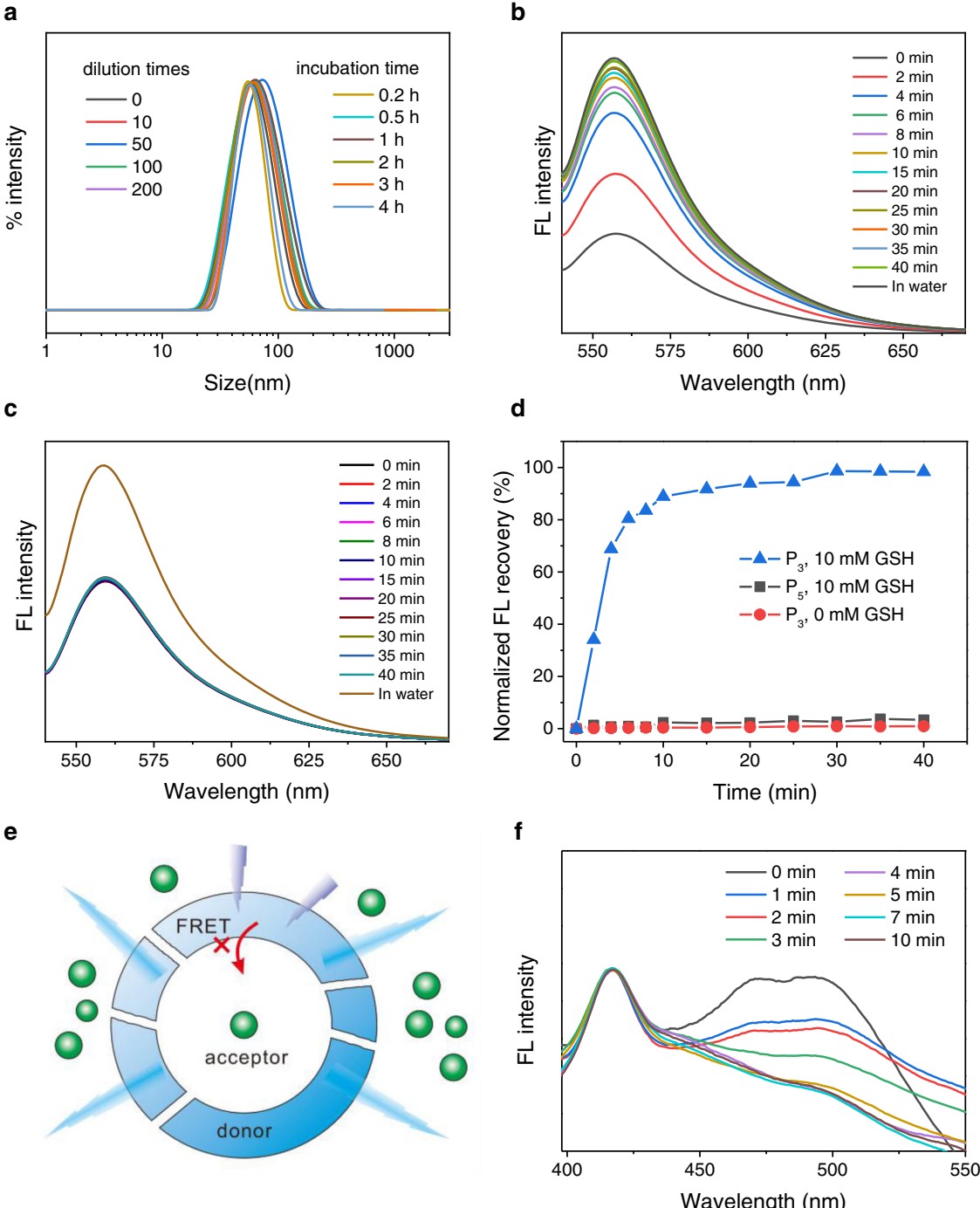

**Fig. 4 | Ultrahigh stability and superfast responsivity. a** Size distribution of $P_3$ assemblies diluted with water for different times and those incubated with SDS (0.02 M) for different times. **b, c** Fluorescence spectra ($\lambda_{ex}$ = 526 nm) of R6G-loaded $P_3$ (**b**) and $P_5$ assemblies (**c**) incubated with 10 mM of GSH for different times. **d** Release profiles of R6G from PU assemblies in the media with or without 10 mM of GSH. **e** Schematic illustration of drug release monitored by the FRET effect between the donor (PU assemblies) and the encapsulated acceptor (QD). **f** Fluorescence spectra of QD-loaded PU assemblies incubated with 10 mM of GSH for different times.

nuclei were 73%, 56%, and 13%, respectively, for $P_4$, $P_2$, and non-responsive $P_5$ (Fig. 5g). As a result, the $P_4$ nanosystem exhibited most potent drug efficacy against cancer cells, with a median inhibitory concentration (1.3 µg mL$^{-1}$) much lower than those of other polymeric formulations (1.8–6.9 µg mL$^{-1}$) (Supplementary Fig. 39, Supplementary Table 3). Furthermore, the drug-free assemblies did not exhibit any inhibitory effect against L929 cells, suggesting good cytocompatibility of PUs (Supplementary Fig. 40).

## In vivo animal studies
To evaluate the antitumor treatment effect of PUs in vivo, taking $P_2$, $P_4$, and $P_5$ as examples, the DOX-loaded nanovehicles were intravenously administered into nude mice bearing MCF-7 tumors. The ultrahigh stability and superfast responsivity of PUs enabled a relatively long circulation time and tumor-specific accumulation of DOX, as evidenced by in vivo and ex vivo imaging (Fig. 6a, b, Supplementary Fig. 41). The mice injected with DOX@PUs did not show apparent loss of body

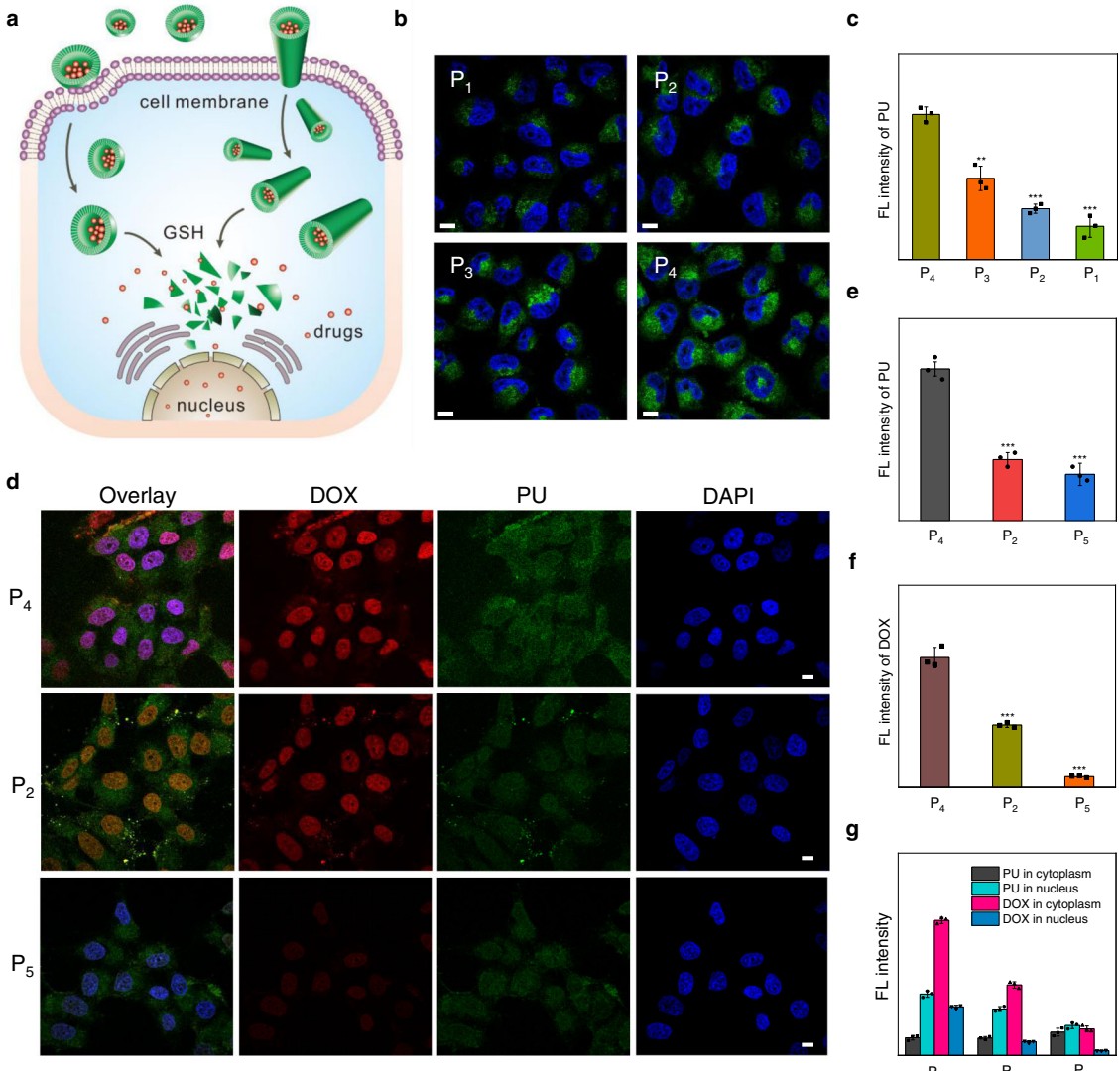

**Fig. 5 | Intracellular drug delivery. a** Schematic illustration of cellular uptake and intracellular drug delivery properties of PU assemblies with different conformations and morphologies. **b** CLSM images of MCF-7 tumor cells incubated with PU assemblies for 2 h. Nuclei of cells were stained with 2-(4-amidinophenyl)-6-indolecarbamidine dihydrochloride (DAPI, blue). The fluorescence of PU channel was green, scale bars: 5 μm. **c** Fluorescence intensity of PUs in MCF-7 cells after 2 h of incubation. Data are presented as mean ± SD ($n$ = 3 independent cells, left to right: **$p$ = 0.0084, ***$p$ = 0.0009, ***$p$ = 0.0007). **d** CLSM images of MCF-7 tumor cells incubated with DOX-loaded PU assemblies for 2 h. Nuclei of cells were stained with DAPI (blue), scale bars: 5 μm. **e-g** Mean fluorescence intensity of PUs (**e**), DOX (**f**), and normalized fluorescence intensity in the cytoplasm and nucleus (**g**). Data are presented as mean ± SD ($n$ = 3 independent cells, **e**, left to right: ***$p$ = 0.0009, ***$p$ = 0.0008. **f**, left to right: ***$p$ = 0.0009, ***$p$ = 0.0003). One-way ANOVA with a Tukey post hoc test was used to establish statistical significance for **c**, **e**, and **f**. Statistical significance: *$p$ < 0.05; **$p$ < 0.01; ***$p$ < 0.001.

weight during the days of injection compared with those treated with free DOX (Supplementary Fig. 42), implying reduced systemic toxicity of polymeric formulations. The tumor growth was inhibited greatly by the treatments of different formulations. In particular, $P_4$ nanocarriers with sheet-like structure and higher content of disulfide linkages exhibited superior antitumor effect as compared with other groups (Fig. 6c). The tumor weight inhibition (TWI) of $P_4$ (97.3%) was much higher than those for $P_2$ (61.3%), $P_5$ (55%), and free DOX (34%) (Fig. 6d). Furthermore, histological analyses with hematoxylin and eosin (H&E) staining, nuclear-associated antigen (Ki67) and terminal deoxynucleotidyl transferased dUTP nick end labeling (TUNEL) revealed that the mice receiving $P_4$ treatment showed most severe cell remission and necrosis, the lowest activity of cell proliferation and the highest percentages of apoptotic tumor cells (Fig. 6e). In addition, CLSM images of tumor slices further confirm that $P_4$ group showed more serious damages of cancer cells than other groups, which was well visualized by remarkably stronger DOX and polymeric autofluorescence (Fig. 6f). The

results validate the high potential of PUs as a smart carrier for targeted drug delivery and effective cancer treatment[70]. In addition, these bioresponsive polymers, constructed from natural amino acid derivatives, are biocompatible, structurally simple, and easy to clinically translate, which will open up many exciting opportunities in the field of biomedical applications.

In summary, we designed and synthesized a class of novel amino acid-constructed and PU-based peptidomimetics via facile chemistry. The polymers exhibited peptide-like ordered secondary conformations and fluorescence protein-mimic AIE emissions. Upon manipulation of H-bond by external media, the polymeric assemblies exhibited interesting conformational transitions, reversible vesicular size expanding and shrinking phenomenon and nanotube-to-vesicle deformation, accompanied by change of fluorescent colors. In addition, the nanovessels possessed ultrahigh stability and superfast responsive property within tumor cells, thus allowing for an attractive "on-off" switch for controlled drug delivery and effective cancer

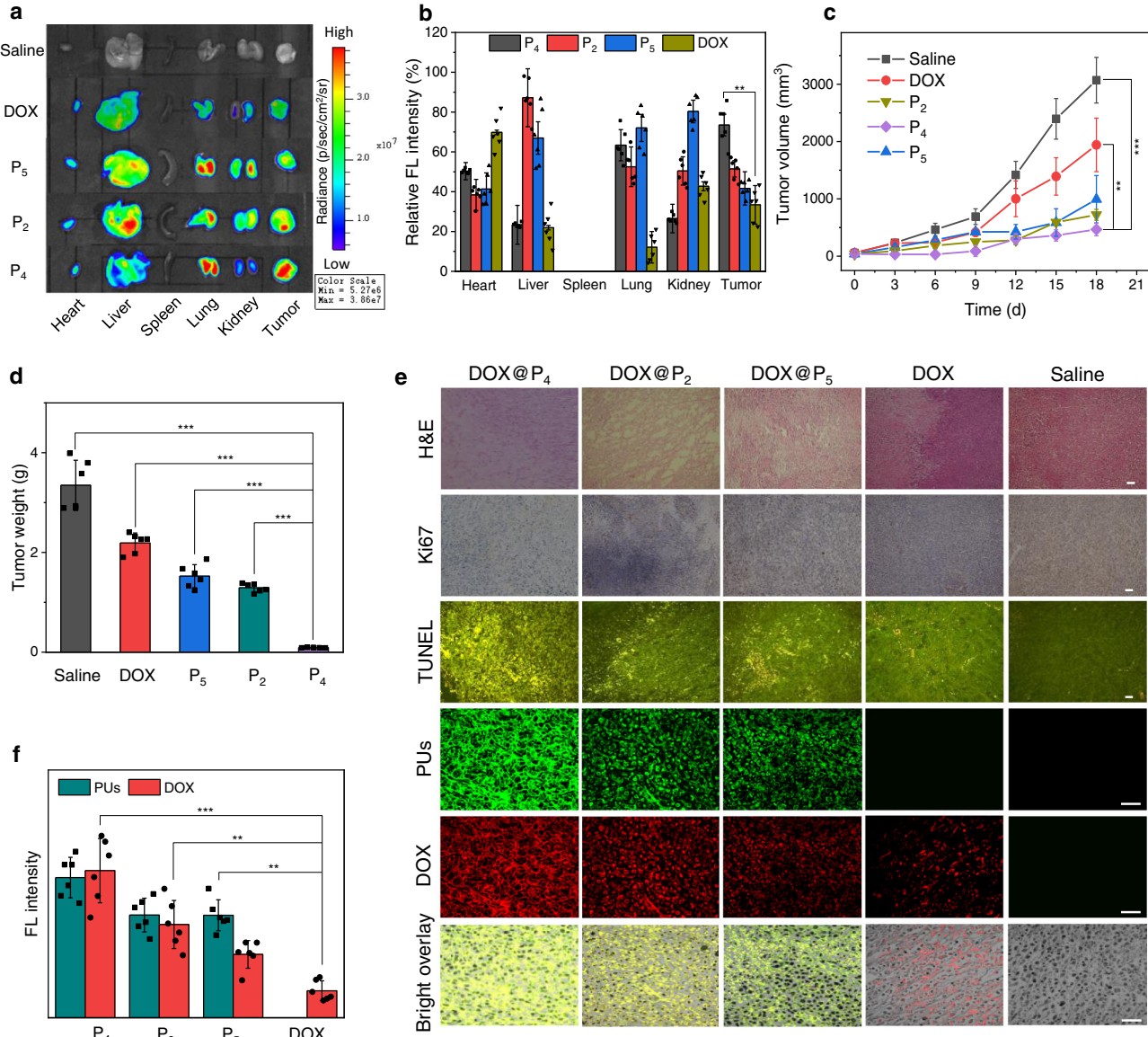

**Fig. 6 | In vivo biodistribution and antitumor efficacy. a** Fluorescence images of major organs and tumors at 24 h after the intravenous administration of saline, free DOX or DOX-loaded PU assemblies ($\lambda_{ex}$ = 480 nm, $\lambda_{em}$ = 600 nm). The experiments were repeated independently three times. **b** Semi-quantitative analysis of DOX fluorescence in major organs and tumors. Data are presented as the mean ± SD ($n$ = 6 independent animals, **$p$ = 0.0061). **c** The growth curves of subcutaneous MCF-7 tumors after intravenous injection of saline, free DOX, DOX@$P_2$, DOX @$P_4$ and DOX@$P_5$ (DOX-equiv. dose, 5.0 mg kg$^{-1}$, $n$ = 6 independent animals per group, left to right: **$p$ = 0.0065, ***$p$ = 0.0008). **d** The average tumor weight of each group at the experimental endpoint ($n$ = 6 independent animals, left to right: ***$p$ = 0.0001, ***$p$ = 0.0003, ***$p$ = 0.0005, ***$p$ = 0.0007). **e** Representative histological features and CLSM images of the tumor sections stained with H&E, Ki67, and TUNEL, scale bars: 100 μm. **f** Mean fluorescence intensity of PUs and DOX in the tumor sections. Data are presented as the mean ± SD ($n$ = 6 independent animals per group, left to right: ***$p$ = 0.0007, **$p$ = 0.0055, **$p$ = 0.0087). One-way ANOVA with a Tukey post hoc test was used to establish statistical significance for **b**, **c**, **d**, and **f**. Statistical significance: *$p$ < 0.05; **$p$ < 0.01; ***$p$ < 0.001.

therapy in vivo. Our work would help understand the folding and hierarchical assembly of biological macromolecules, and offer a new class of protein-mimic and nontypical fluorescent polymeric materials for biosensing, drug delivery, and theranosis applications.

## Methods

### General procedure for the synthesis of PUs

L-Cys·OMe·2HCl, D-Cys·OMe·2HCl (3.4–6.8 g), or L-Lys·OEt·2HCl (3.7 g) was dissolved in *N,N*-dimethylacetamide (DMAC, 40 mL) and injected into an argon-protected reaction flask, and triethylamine (TEA, 6 mL) was added at the same time. Then a DMAC solution of LDI (2.4–4.7 g) was dropped into the reaction flask, and the reaction was carried out at 60 °C for 2 h. Thereafter, stannous octanoate and mPEG5000 (5.5–11 g)

were added into the system and reacted at 80 °C for 3 d. The resultant solution was transferred into a dialysis bag (MWCO 6000) and dialyzed against deionized water for 3 d with deionized water changed once 3 h. Then the solution was lyophilized to obtain a solid (80–90% yield).

### Circular dichroism

CD was analyzed on a J-1500-150 spectrometer (JASCO Corporation, Japan) at room temperature in the range of 190–300 nm. The molar ellipticity [$\theta$] was calculated by the following equation: [$\theta$] = ($\theta$ × 100 × $M_w$)/($C$ × $l$), where '$\theta$' was obtained from CD spectrometer, $M_w$ is the molecular weight of amino acid residue, '$C$' is the concentration of sample, and '$l$' is the width of cuvette.

## AIE properties of PUs

Fluorescence spectra ($\lambda_{ex} = 365$ nm) of PUs dispersed in mixtures of DMF-CHCl$_3$ (1 mg mL$^{-1}$) with different CHCl$_3$ fractions ($f_w$) were recorded on an F-4600 FL spectrophotometer (Hitachi, Ltd., Japan) at 25 °C, with an emission slit width of 5 nm and constant scan rate. The emission images of the mixture solutions were captured on a Leica microsystem (DMi 8).

## ThT fluorescence assay

ThT (0.1 mg mL$^{-1}$) was added to PU assemblies (1 mg mL$^{-1}$) and the fluorescence measurement was conducted on an F-4600 FL spectrophotometer (Hitachi, Ltd., Japan) with a bandwidth of 5 nm and $\lambda_{ex}$ of 440 nm.

## Responsiveness of PUs

R6G-loaded PUs assemblies were incubated in a 10 mM GSH solution, recording the fluorescence emission spectra overtime at $\lambda_{ex}$ of 526 nm on an F-4600 FL spectrophotometer (Hitachi, Ltd., Japan). The FL intensity recover rate was calculated by $(I - I_0)/(I_w - I_0) \times 100\%$ as a function of time, where $I$ is the FL intensity of R6G at different times, $I_0$ is the FL intensity at initial time, $I_w$ is the FL intensity of free fluorescent probe dissolved in water with the same concentration as that encapsulated in assembled solutions. For FRET measurement, QD-loaded PUs assemblies were treated with 10 mM GSH and measured with an F-4600 FL spectrophotometer (Hitachi, Ltd., Japan) at different time points. The emission spectra were collected from 440 to 700 nm at a $\lambda_{ex}$ of 430 nm.

## Confocal laser scanning microscope (CLSM)

P$_1$, P$_2$, P$_3$, and P$_4$ assemblies and DOX-encapsulated assemblies (1 mg mL$^{-1}$) were added to the surface of a glass slide and sealed with cover glass, and then maintained at 4 °C for 12 h. The samples were imaged on a confocal laser scanning microscope (CLSM, Olympus FV1000, Nikon A1RMP, Japan).

## Cell internalization and intracellular delivery

MCF-7 cells were purchased from West China Medical Center of Sichuan University and cultured with Dulbecco's modified Eagle's medium (DMEM) medium (Hyclone) containing 10% fetal bovine serum (Gibco) in an atmosphere of 5% CO$_2$ at 37 °C. The cells were seeded in a six-well plate (a coverslip was placed in every well before use) at a density of $1 \times 10^5$ cells per well and cultured overnight. Then blank and DOX-loaded PU assemblies were added separately into the plate and incubated for 2 h. Next, the medium was removed and the plate was washed with PBS for three times. Then the cells were fixed with 4% formaldehyde for 30 min and stained with DAPI for 10 min. At last, the coverslips were mounted with 50% glycerol solution and observed on a CLSM.

## Antitumor treatment

MCF-7 tumor-bearing nude mice were divided into five groups (six mice per group). When the tumors had grown to 30–50 mm$^3$, the nude mice were administrated with DOX@P$_5$, DOX@P$_2$, and DOX@P$_4$ via tail vein every 3 d for 15 d at a DOX dose of 5 mg kg$^{-1}$. Mice injected with free DOX and saline were set as positive and negative controls, respectively. The tumor sizes were recorded every 3 d using a digital caliper. On the day of 18, all the mice were sacrificed and the tumors were excised and weighed. The major organs (heart, liver, spleen, lung, and kidney) and tumors were collected, embedded with paraffin, and cut into 5-μm-thick sections. The sections were observed with CLSM to assess the damages of tumor tissues and distribution of drugs and polymers, and subjected to H&E staining, TUNEL assay, and Ki67 immunohistochemistry analysis.

## Statistics and reproducibility

The quantitative data obtained were presented as means ± standard deviations (SD). Statistical analysis was performed using the GraphPad Prism (version 8.0.2). One-way analysis of variance (ANOVA) with a Tukey post hoc test was performed to determine the statistical significance within the data at 95% confidence levels ($P < 0.05$). Micrographs, CLSM, SEM, and TEM experiments were repeated three times independently with similar results, and typical images are shown.

## Reporting summary

Further information on research design is available in the Nature Research Reporting Summary linked to this article.

## Data availability

Source data is available for Figs. 1–6 and Supplementary Figs. 3–11, 13, 15–19, 21–25, 28–29, 31–37, 39–40, and 42 in the associated source data file. The data that support the findings of this study are available within the paper, Supplementary Information files and Source data file, or are available from the corresponding author upon request.

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

## Acknowledgements
We thank Prof. Yong Luo and Shanling Wang at Sichuan University for help with TEM observation. This work was supported by the National Natural Science Foundation of China (52022062, M.D., 51873118, M.D., 51733005, H.T., 21474064, M.D.), the Application and Basic Research of Sichuan Department of Science and Technology (2021YJ0225, M.D.) and the Project of State Key Laboratory of Polymer Materials Engineering.

## Author contributions
Y. Zhou, F.F., and M.D. conceived and designed the research. Y. Zhou, F.F., J.Z., and R.W. synthesized and characterized the materials. Y. Zheng and C.P. performed the fluorescent studies and data analysis. Y. Zheng, H.L., and Z.W. performed the in vitro and in vivo studies. Y. Zhou and M.D. wrote the manuscript with feedback from all the authors. M.D. devised the study and supervised the project with J.L., H.T., and Q.F. All authors discussed the results and commented on the manuscript.

## Competing interests
The authors declare no competing interests.
