## [Peer Review File · Nature Communications]

Intrinsically Fluorescent Polyureas toward Conformation-Assisted Metamorphosis, Discoloration and Intracellular Drug DeliveryREVIEWER COMMENTS

Reviewer #1 (Remarks to the Author):

This manuscript reports an interesting approach towards macromolecular peptidomimetics (and not pseudo-polypeptides) with non-conventional luminescent properties. Nevertheless, the presentation of experimental data and discussions lack of scientific rigor and should be carefully reconsidered. As it is, some points are unconvincing and unsupported. Overall, the manuscript may represent a significant step further for the scientific community in the preparation and application of macromolecular peptidomimetics. Hence, I could only recommend the publication of this manuscript in Nature Communications if the following considerations are taken into account.

1) Biblio. The introduction is too vague, somehow incorrect and lacks of precisions. Some important points to consider:

- a. The luminescence phenomenon described by the authors has already been observed with this class of polymer (for instance, *Mol. Syst. Des. Eng.* 2018, 3, 364-375).
- b. The term pseudo-polypeptide is not appropriate and should be replaced by peptidomimetic. In this context, and in contrast with the authors' statement, peptidomimetic polymers containing amino acids linked by amide bonds exist: these are called polypeptoids (for instance, *J. Am. Chem. Soc.* 2021, 143, 3697).
- c. The analogy with proteins and their fluorescence properties is difficult to understand. In nature, these proteins use fluorophores which is not the case in this work. The title and introduction should be revised to minimize the analogy with these natural systems.
- d. The properties of PUs and their stabilized conformations have already been extensively studied (For instance *J. Mat. Sci.*, 2014, 49, 7339). The term "protein-like functions" is too vague in this context and the following sentence is therefore untrue: "to our knowledge, PUs with ordered conformations and protein-like functions has not been reported so far."
- e. The context of oligoureas in the field of foldamers must also be mentioned, especially considering some key works concerning their ordered conformation and fluorescence (Gilles Guichard among other possible authors).

2) Molecular design and characteristics.

- a) Many details and characterizations are missing in this part. For example, schema 1 and 2 in esi should be more detailed, figure 1, 2, 3 in esi contains unassigned peaks, solvents are missing in some procedures or in the description of figure 6, yields in procedures etc. For this same figure 6, it would be interesting to provide the complete SEC chromatogram. On the other hand, on this figure 6, how do the authors explain the low dispersity obtained? Was it correctly measured? The results are questionable because polycondensations give dispersity values around 1.5 or above. Values below 1.2 are obtained with controlled polymerization processes.
- b) Concerning the chemical structure, there are already articles on PUs made from Lysine. The use of cysteine is original but generates links that are sensitive to redox conditions. How do the authors monitor the integrity of these bonds? Are they not at the origin of the modifications/properties observed in the nanomaterials? Still concerning this chemical structure, I do not understand the scheme depicting ethoxy or methoxy pendant chains. There must be errors in these schemes considering the characterizations that are provided.
- c) I am not completely convinced by the fact that the authors only get triblocs. What evidence precludes the presence of diblocks or non-pegylated copolymers? On the other hand, figure 1a, it is incorrect to use the letters that are usually assigned in peptides: cysteine for example corresponds to a monomeric unit combining two cysteines. This scheme needs to be revised.

3) Self-assembly of PUs. The self-assembly section contains many weak points and assertions that require rigorous review.

- a) The first and very important concern is about the images obtained by TEM. The images are not representative and are often blurred (and poorly contrasted). Representative means that a significant amount of object must appear on the image (or in an additional image given in the east). The formation of nanotubes is therefore poorly verified and characterized and without additional data, it is

a statement to be removed from this manuscript (CLSM images are even less convincing).
b) Looking carefully at Figure 1 b and c, the P4 sample contains mainly small objects (more than 99%). The intensity in DLS varies with the power 6 of the size. In number, objects at 100 nm are therefore traces in the medium. This is a second point showing that nanotubes are not formed after self-assembly of P4 (and instead small micelles certainly). Still concerning this size of nano-objects but in figure 1e, what happens for P3 and P4 nano-objects at higher TFA concentrations?
c) Figure 1h, the claim that the process can be implemented multiple times is not experimentally supported. The authors should add these characterizations.

4) Molecular conformation.

a) The term beta sheet conformation is somehow not appropriate as evidenced in fig 2a (for instance, the positive peak at 200 nm is missing). It would be better to use sheet-like conformation or associate the conformational state to the creation of an intermolecular hydrogen bonding network.
b) Instead of using TFA, what happens if we add TFE or methanol? In both cases, the hydrogen bonds are strengthened with polypeptides (see Tim Deming's work for instance). On the other hand, is there an impact of the chirality of the amino acids (racemic mixture versus homochirality) on the conformation (or on the luminescent properties)?
c) The following statement is wrong: "the results demonstrate for the first time a conformation-directed morphological transition in non-peptide polymers" (See for instance the work of Jing Sun on polypeptoids).

5) Nonconventional fluorescence properties. The major point to review here is the scientific context. There is no real analogy with natural proteins from jellyfish. In the latter, the proteins have properties related to the presence of a fluorophore. The introduction of this paragraph should therefore be revised considering that this principle of luminescence is already well documented and that key references are missing (see my comments in the introduction).

Minor comments:

- Table 1 from the esi should appear in the main manuscript (with correct SEC values).
- In the intro., hydrophobic interactions do not exist (this is an entropic effect).
- Ref2-8 are not necessarily well chosen (see polypeptide polymers and secondary structures)
- Fig 1d, the label in water is confusing. This should be replaced by "dye alone".
- Fig 4a: which PU is it?

Reviewer #2 (Remarks to the Author):

In the manuscript, Zhou et al. described a unique urea-constructed pseudo-poly(amino acid), which exhibited peptide-like ordered secondary conformations and fluorescence protein-inspired nonconventional autofluorescence with typical aggregate induced emission (AIE) character. Particularly interesting is the reversible vesicular size expanding and shrinking phenomenon and nanotube-to-vesicle deformation of the assemblies accompanied by an exciting change of fluorescent colors. Moreover, the nanovesicles possessed ultrahigh stability, superfast responsive property, and conformation-assisted cell internalization efficiency. The cell uptake was visualized by the autofluorescence of the polymers, and the drug release profiles were monitored by a fluorescence resonance energy transfer (FRET) phenomenon between the polymers and encapsulated dyes. In an overall view, this manuscript was very appealing on reading because the authors went all the way from polymer design and synthesis, characterization of the assembly, analysis of secondary conformations and fluorescence properties, and performed animal studies to show the use of these materials as an on-off switchable nanocarrier for tumor therapy in vivo. The conclusions were well supported by numerous analytical methods the authors have employed. Thus, the reviewers would

recommend the acceptance of this manuscript for publication after some minor revisions.

1. According to supplementary information, P5 was synthesized using LDI and Lys•OMe•2HCl monomers. In Supplementary Table 1, however, P5 consists of only LDI residues. This is confusing and needs some explanation.
2. The size changes of all the polymer assemblies treated with trifluoroacetic acid (TFA) were presented in Figure 1e, but the change in size distributions of P4 was not shown in Figure 1f. Moreover, the volume fractions of TFA added were not specified.
3. Could the theory and characterization methods about the secondary conformations of poly(amino acid)s be applied to non-peptide polymers?
4. The fluorescence behaviors of the polymers were fascinating. What about the fluorescence quantum yield of the polymers?
5. In the animal study, the change of the body weights of the mice should be provided.
6. In addition, the scale bars should be added in the confocal laser scanning microscope (CLSM) images.
7. There are a couple of minor grammar or type errors in the manuscript, for instance “Th T” should be “ThT” in Figure 2b.
8. Some typos in the reference list should be corrected, and the recently published review or research articles should be discussed in the revision, for example, [10.15212/bioi-2021-0016](https://doi.org/10.15212/bioi-2021-0016), [10.1016/j.nantod.2019.100800](https://doi.org/10.1016/j.nantod.2019.100800).

Please find below a summary of our point-by-point response to the reviewer comments.

To Reviewer #1

1) Comment: Biblio. The introduction is too vague, somehow incorrect and lacks of precisions. Some important points to consider:

a. The luminescence phenomenon described by the authors has already been observed with this class of polymer (for instance, *Mol. Syst. Des. Eng.* 2018, 3, 364-375).

Response: We acknowledge the reviewer's critical comment. We have deeply revised the introduction section to make it clearer, and corrected some inappropriate or inaccurate statements.

We agree with the reviewer that the luminescence phenomenon of nonconventional chromophore has been widely reported recently (*Mol. Syst. Des. Eng.* 2018, 3, 364-375; *Polym. Chem.* 2017, 8, 1722-1727; *J. Polym. Sci. A Polym. Chem.* 2017, 55, 560-574; *Prog. Polym. Sci.* 2019, 90, 35-117; *Mater. Chem. Front.* 2021, 5, 60-75; *Chem. Soc. Rev.* 2021, 50, 12616-12655). To address the comment, **we have described and reviewed these studies** in the revised manuscript: “*Nonaromatic polyurethanes, polypeptides and other aliphatic compounds have been shown to generate noticeable intrinsic fluorescence under suitable conditions based on a clustering-triggered emission (CTE) mechanism (Polym. Chem. 2017, 8, 1722-1727; J. Polym. Sci., Part A: Polym. Chem. 2017, 55, 560-574; Mol. Syst. Des. Eng. 2018, 3, 364-375; Prog. Polym. Sci. 2019, 90, 35-117; Mater. Chem. Front. 2021, 5, 60-75; Chem. Soc. Rev. 2021, 50, 12616-12655). However, most of these polymers were fluorescent when concentrated or aggregated as powders and films but nonemissive in dilute solutions (Mol. Syst. Des. Eng. 2018, 3, 364-375; Small 2016, 12, 6586-6592). In this work, the compact H-bonding of PUs resulted in the close proximity of carbonyl moieties to form a “heterodox cluster”, which gave rise to emission under UV irradiation (Fig. 3a) (Macromolecules 2015, 48, 64-71). Of interest, the dilute PU solutions, assemblies and films emitted cyan blue and green fluorescence in the*

presence of UV irradiation” (Page 15, Lines 6-15); “*It should be noted that Bonifacio et al. has observed pH-dependent weak fluorescence in polyurea dendrimers, and the light emission originated from the protonation of primary and tertiary amino groups (Angew. Chem. Int. Ed. 2012, 124, 5252-5255).*” (Page 16, Lines 2-5).

b. Comment: The term pseudo-polypeptide is not appropriate and should be replaced by peptidomimetic. In this context, and in contrast with the authors’ statement, peptidomimetic polymers containing amino acids linked by amide bonds exist: these are called polypeptoids (for instance, *J. Am. Chem. Soc.* 2021, 143, 3697).

Response: Thanks for the helpful suggestion. The term **peptidomimetic** is in a broad sense, referring to any sequence designed to mimic a peptide structure and/or function, but whose backbone is not solely based on α -amino acids (*J. Am. Chem. Soc.* 1996, 118, 11, 2539–2544; *Cell. Mol. Life Sci.* 2011, 68, 2255–2266). **Pseudopeptide** is a peptidomimetic where one or more peptide bonds have been replaced with an isostere (*Curr. Pharm. Des.* 2010, 16, 3185-3203). **Peptoids** are peptidomimetic polymers composed of repeating *N*-substituted glycine monomer units, where the side chain is appended to the nitrogen atom rather than the α -carbon (*Acc. Chem. Res.* 2016, 49, 3, 379–389). In general, the peptoid backbone is inherently flexible due to the lack of main-chain chirality and hydrogen bond donor (*ACS Nano* 2013, 7, 6, 4715–4732; *Macromolecules* 2013, 46, 20, 8213–8223). In our work, the natural amino acid-derived PUs should belong to a class of peptidomimetics, which is significant different from peptoids that do not have chirality and H-bond donor in the backbone. The PUs could also be called **pseudo-poly(amino acid)s (PAAs)**, which in their structures, naturally occurring amino acids are connected by non-amide bonds such as ester, carbonate, urethane, and imide bonds (*Colloid Polym. Sci.* 2011, 289, 1055–1064; *Polym. Mater. Sci. Eng.* 1984, 51,119–121; *J. Am. Chem. Soc.* 1987, 109, 817-820; *Biomaterials* 1996, 17, 463-468). According to the reviewer’s suggestion, **we have deleted the term pseudo-polypeptide** from the manuscript, and **replaced it with pseudo-poly(amino acid)s (pseudo-PAAs) or peptidomimetics** (Page 1, Line 8; Page 2,

Lines 10 and 15; Page 3, Line 8; Page 20, Line 2; Page 13, Line 19; Page 21, Line 21).

c. Comment: The analogy with proteins and their fluorescence properties is difficult to understand. In nature, these proteins use fluorophores which is not the case in this work. The title and introduction should be revised to minimize the analogy with these natural systems.

Response: Now **we have revised throughout the manuscript** (including title and introduction) and **minimized the analogy with natural fluorescent proteins** (Page 1, Line 1; Page 1, Line 21; Page 2, Line 4; Page 3, Lines 9-10; Page 3, Line 12; Page 15, Lines 6-11).

d. Comment: The properties of PUs and their stabilized conformations have already been extensively studied (For instance *J. Mat. Sci.*, 2014, 49, 7339). The term “protein-like functions” is too vague in this context and the following sentence is therefore untrue: “to our knowledge, PUs with ordered conformations and protein-like functions has not been reported so far.”

Response: We appreciate the critical comment. It should be note that the research (*J. Mat. Sci.*, 2014, 49, 7339) focused mainly on the effect of hard segment structure and chemistry, and the hydrogen bonding on the microphase-separated morphology and properties of polyurethanes and polyurea elastomers, which did not mention helical- or sheet-like ordered secondary structures. However, after careful literature search, we did find that polyurethanes and oligoureas with stabilized conformations have been already reported (*Polymer* 2011, 52, 3745-3751; *J. Am. Chem. Soc.* 2005, 127, 2156–2164; *Angew. Chem. Intern. Ed.* 2010, 49, 1067-1070; *J. Am. Chem. Soc.* 2013, 135, 4884–4892). **We have cited the references and described these work** in the revised manuscript: “*In fact, sequence-defined oligoureas have been established as attractive peptidomimetic foldamers that adopt robust helical conformation stabilized by three-center H-bonds* (*J. Am. Chem. Soc.* 2005, 127, 2156–2164; *Angew. Chem. Intern. Ed.* 2010, 49, 1067-1070; *J. Am. Chem. Soc.* 2013, 135, 4884–4892)” (Page 3, Lines 3-6);

“we wonder whether these polymers possess different secondary conformations, as chiral tyrosine-constructed polyurethanes have shown helical configurations (Polymer 2011, 52, 3745-3751)” (Page 11, Lines 1-3). Furthermore, as suggested by the reviewer, **we have deleted the vague term “protein-like functions” and replaced the following sentence** *“to our knowledge, PUs with ordered conformations and protein-like functions has not been reported so far”* **with** *“Nonetheless, PU-based functional pseudo-PAA s with ordered secondary structures and conformation-mediated functionalities have been rarely reported”* (Page 3, Lines 7-9).

e. Comment: The context of oligoureas in the field of foldamers must also be mentioned, especially considering some key works concerning their ordered conformation and fluorescence (Gilles Guichard among other possible authors).

Response: Thanks for the constructive comment. **We have included the context of oligoureas in the field of foldamers** in the revised introduction section and **cited the relevant literature**: *“In fact, sequence-defined oligoureas have been established as attractive peptidomimetic foldamers that adopt robust helical conformation stabilized by three-center H-bonds (J. Am. Chem. Soc. 2005, 127, 2156–2164; Angew. Chem. Intern. Ed. 2010, 49, 1067-1070; J. Am. Chem. Soc. 2013, 135, 4884–4892)”* (Page 3, Lines 3-6).

2) Molecular design and characteristics.

a. Comment: Many details and characterizations are missing in this part. For example, schema 1 and 2 in esi should be more detailed, figure 1, 2, 3 in esi contains unassigned peaks, solvents are missing in some procedures or in the description of figure 6, yields in procedures etc. For this same figure 6, it would be interesting to provide the complete SEC chromatogram. On the other hand, on this figure 6, how do the authors explain the low dispersity obtained? Was it correctly measured? The results are questionable because polycondensations give dispersity values around 1.5 or above. Values below 1.2 are obtained with controlled polymerization processes.

Response: According to the thoughtful suggestions, **we have added more details** in the supplementary Scheme 1 and 2, and **carefully checked and reassigned all the peaks** in supplementary Figs. 1-3. Moreover, we have **provided the solvents for polymer synthesis and GPC determination as well as the yields of PUs** in both the figure captions (Supplementary Scheme 1 and 2, Fig. 6) and experimental sections (Page 24, Line 9 in revised manuscript; Page 2, Lines 37-40 in Supplementary information). As suggested by the reviewer, **the complete SEC chromatograms have been provided** in supplementary Fig. 6. In addition, the number-average and weight-average molecular weights were measured by GPC in DMF/LiBr (2 g L⁻¹) with polymethyl methacrylate (PMMA) standard. The polydispersity index (\mathcal{D}) was calculated by the ratio between the weight average and number average molecular weights (M_w / M_n) and listed in Table 1. The \mathcal{D} values were in the range of 1.02-1.65, which are relatively low because of the less reaction components, high purity of monomers, high reactivity of primary amine groups with isocyanate groups under mild conditions, as well as over thirty-years of systematic experience in polyurethane/urea preparation and characterization in our laboratories. In fact, we have even synthesized multiblock polyurethanes via the polycondensation of 5-8 components and these copolymers also showed narrow molecular weight distribution (\mathcal{D} 1.10-1.66) (*Adv. Mater.* 2012, 24, 3639-3645; *ACS Nano* 2013, 7, 1918-1928). Moreover, the preparation of polyurethanes with dispersity values around 1.45 or below has also been widely reported in the literature (*Macromolecules* 2020, 53, 5992–6001; *Eur. Polym. J.* 2013, 49, 823-833; *Macromolecules* 2020, 53, 3349–3357). To address the comment, **we have added the method for measuring dispersity** in Table 1. Moreover, to further confirm the triblock structures of PUs with narrow molecular weight distributions, **we have synthesized a control polymer containing diblock and triblock mixtures** (DTPU) by feeding half amount of mPEG. As expected, DTPU exhibited much larger polydispersity (\mathcal{D} 1.92), bimodal molecular weight distribution, and reduced integral area of PEG in the ¹H NMR spectrum. The additional results were added in the revised supporting information (Supplementary Figs. 7 and 8) and discussed in the revised manuscript

(Page 4, Lines 13-17).

b. Comment: Concerning the chemical structure, there are already articles on PUs made from Lysine. The use of cysteine is original but generates links that are sensitive to redox conditions. How do the authors monitor the integrity of these bonds? Are they not at the origin of the modifications/properties observed in the nanomaterials? Still concerning this chemical structure, I do not understand the scheme depicting ethoxy or methoxy pendant chains. There must be errors in these schemes considering the characterizations that are provided.

Response: Thanks for the critical comment. It is true that there are a variety of polyurethanes made from lysine derivatives (*J. Biomater. Sci. Polymer. Ed.* 2001, 12, 851–873; *Biomaterials* 2000, 21, 1247-1258; *Polymer* 2006, 47, 785-798), including those reported in our previous work (*Biomacromolecules* 2009, 10, 2857–2865; *Soft Matter* 2010, 6, 2087–2092; *Biomaterials* 2011, 32, 9515-9524; *Adv. Mater.* 2012, 24, 3639-3645; *ACS Nano* 2013, 7, 1918-1928; *Biomaterials* 2017, 145, 138-153; *Adv. Sci.* 2020, 7, 1902701). However, most of these studies used amino acid-based monomers for improving the biocompatibility of materials or minimizing the toxicity of polymeric degradation products. While in this work, we incorporated L-lysine ethyl ester and L-cystine dimethyl derivatives into polyureas mainly for four reasons: (1) the bifunctional monomers could quickly form urea-linked pseudo-poly(amino acid)s with regular structures under mild conditions; (2) the molecular design enabled interesting H-bonding and carbonyl-carbonyl $n \rightarrow \pi^*$ interactions among the urea linkages in the backbone and ester carbonyl groups in the side chains, which gave rise to different secondary conformations (such as helical and sheet-like structures); (3) the urea-clustered carbonyl moieties allowed an aggregation-induced emission under UV irradiation; (4) the high content of disulfide linkages within the polymeric backbone enabled a high responsivity to intracellular level of GSH (10 mM), resulting in a superfast and nearly completed release of payloads, which is much faster than those reported for other redox-responsive drug delivery systems (*Angew. Chem. Int. Ed.* 2017,

56, 9603-9607; *Angew. Chem.* 2015, 127, 9350-9355).

It should be noted that we used **cystine** rather than **cysteine** in our work, the latter contains a pendant thiol group and is very sensitive to oxidation, while the former bears disulfide linkages that is relatively stable under normal physiological conditions but cleaved in response to intracellular level of GSH. The stimuli-responsive property of cystine-constructed PUs has been confirmed by FRET and drug release experiments (Fig. 3). According to the comment, **we have performed additional experiment to monitor the integrity of these polymers during experiments** taking cystine-bearing P₃ as an example. In particular, P₃ was incubated with TFA for one week and collected for NMR test. The results proved that the disulfide linkages and other characteristic peaks remained unchanged before and after treatment, demonstrating the integrity of these bonds. **The additional experimental procedures and results have been provided** in the revised Supplementary information (Page 5, Lines 36-40; Supplementary Fig. 14) **and discussed** in both the revised manuscript: “*Note that the TFA treatment did not the disrupt the molecular integrity of PUs (Supplementary Fig. 14)*” (Page 8, Lines 13-14) and revised supporting information: “*In addition, to verify the integrity of PU structures during TFA treatments, the polymers were incubated in TFA for one week and analyzed with ¹H NMR. As found in Supplementary Fig. 14, the characteristic peaks and their integral areas were kept nearly unchanged after incubation with TFA, suggesting that the TFA treatment did not disrupt the molecular integrity of PUs*” (Page 5, Lines 36-40).

In addition, **we have added the structures of ethoxy or methoxy pendant chains in the chemical schemes** (Fig. 1) to make the figure more readable, and **carefully checked and reassigned all the peaks** corresponding to ethoxy or methoxy pendant chains in supplementary Figs. 1-3.

c. Comment: I am not completely convinced by the fact that the authors only get triblocks. What evidence precludes the presence of diblocks or non-pegylated copolymers? On the other hand, figure 1a, it is incorrect to use the letters that are usually assigned in

peptides: cysteine for example corresponds to a monomeric unit combining two cysteines. This scheme needs to be revised.

Response: In this study, the PUs were synthesized using LDI and Cys·OMe (or Lys·OEt) as monomers and mPEG as an end-capping agent. It should be noted that the different components do not copolymerize random and simultaneously, but react sequentially *via* a facile two-stage polymerization (Supplementary Scheme 1), where all the monomer feeding and temperature control were carried out carefully step by step. First, excessive amount of LDI was reacted with Cys·OMe for 2 h at 60°C to prepare prepolymers with two isocyanate end groups. Then excessive amount of mono-functional MPEG was used to terminate the polymers at 80 °C for 3 d. Finally, the obtained polymers were purified to remove any unreacted mPEG. In principle, such a procedure can ensure that both the polymeric chain ends were connected with mPEG segments, that said, the resultant products are triblock copolymers. This can be confirmed by the integration of NMR peaks (Supplementary Figs.1-4) as well as the narrow molecular weight distributions (\bar{M}_w 1.02–1.65) with single elution peaks (Table 1 and Supplementary Fig. 6). Otherwise, the presence of diblock or non-PEGylated copolymers will inevitably increase the molecular weight distribution. “*To verify the triblock architecture of PUs, we synthesized a control polymer containing diblock and triblock mixtures (DTPU) by feeding half amount of mPEG. As expected, DTPU exhibited much larger polydispersity (\bar{M}_w 1.92, Table 1), bimodal molecular weight distribution (Supplementary Fig. 7), and reduced integral area of PEG in the ^1H NMR spectrum (Supplementary Fig. 8)*” (Page 4, Lines 13-17). Moreover, **we also carried out an end-group analysis to preclude the presence of diblocks or non-pegylated copolymers.** “*It is believed that diblock or non-PEGylated PUs possess one or two amine groups on the chain end due to the reaction of isocyanate-terminated prepolymers with water (RSC Adv., 2014, 4, 33520-33529). Therefore, we carried out an end-group analysis taking P_3 and DTPU as examples. The polymers were reacted with FITC and subjected to UV-vis and fluorescence measurements. Apparently, DTPU displayed distinct UV absorption at 490 nm and a fluorescence emission peak at 552 nm (Supplementary Fig.*

9), suggesting that it has been successfully labeled with FITC. In contrast, P_3 did not show noticeable FITC signal (Supplementary Fig. 9), thus confirming the PEGylated triblock structure of PUs that do not have any reactive groups” (Page 4, Lines 18-22; Page 5, Lines 1-4). To address the comment, **the additional results were provided** in Table 1 and Supplementary Figs. 7-9, and discussed in the revised manuscript (Page 4, Lines 13-22; Page 5, Lines 1-4). In addition, **we have revised the scheme in Fig. 1a** as suggested by the reviewer, where the letters usually assigned in peptide have been replaced with the initials of amino acid residues.

3) Self-assembly of PUs. The self-assembly section contains many weak points and assertions that require rigorous review.

a. Comment: The first and very important concern is about the images obtained by TEM. The images are not representative and are often blurred (and poorly contrasted). Representative means that a significant amount of object must appear on the image (or in an additional image given in the east). The formation of nanotubes is therefore poorly verified and characterized and without additional data, it is a statement to be removed from this manuscript (CLSM images are even less convincing).

Response: We are grateful to the critical comment. To better characterize the vesicles and nanotubes, **we have repeated TEM and CLSM measurements, performed additional cryo-SEM analyses and replaced the low-quality images** in Fig. 1c and 2i. Moreover, as suggested by the reviewer, **we have provided TEM and cryo-SEM images containing a significant amount of objects** in Supplementary Figs.10 and 11. As seen from the TEM and cryo-SEM images, the formation of tube-like morphology was evident, “*where a number of open-ended tubular structures with large aspect ratios can be observed (Figure 2i and Supplementary Fig. 11)*” (Page 7, Lines 15-17). To provide more evidence for the nanotube formation, we incorporated hydrophilic rhodamine 6G (R6G) and doxorubicin hydrochlorate (DOX·HCl) within the polymer dispersions. Note that the purpose of CLSM observation was not only to observe the morphology of assemblies, but also to detect the fluorescence signal of hydrophilic dyes

to investigate the host-guest features of different assemblies. Although the sizes of PU assemblies are rather small to be close to the lateral resolution of confocal microscope, the size distributions are relatively wide and the nanoparticles may aggregate during drying process. As a result, distinct “fluorescence dots and lines” could be observed in the CLSM images. The result demonstrates that the PU assemblies can accommodate hydrophilic DOX·HCl, thus confirming the formation of vesicular or tube-like architecture with an aqueous core (*J. Am. Chem. Soc.* 2016, 138, 7508–7511; *J. Am. Chem. Soc.* 2018, 140, 6604-6610; *Angew. Chem. Int. Ed.* 2021, 60, 22529-22536). Despite the relatively low resolution of images, the CLSM fluorescence approach is still promising and informative, which has been widely used to probe micelles, vesicles and other nano-objects (*Langmuir* 2011, 28, 2056–2065; *Langmuir* 2012, 28, 11988–11996; *Adv. Funct. Mater.* 2015, 26, 66-79).

b. Comment: Looking carefully at Figure 1 b and c, the P4 sample contains mainly small objects (more than 99%). The intensity in DLS varies with the power 6 of the size. In number, objects at 100 nm are therefore traces in the medium. This is a second point showing that nanotubes are not formed after self-assembly of P4 (and instead small micelles certainly). Still concerning this size of nano-objects but in figure 1e, what happens for P3 and P4 nano-objects at higher TFA concentrations?

Response: We agree with the reviewer that P₄ samples may contain many objects with small size since the intensity distribution in DLS varies with the power 6 of the size. However, the exact size values and distributions determined by DLS are unlikely to be a quantitatively precise reflection of the true diameters and lengths of nanotubes, because DLS method handles anisotropic rod-like particles as spherical particles within a hydrodynamic approximation (*Macromol. Rapid Commun.* 2011, 32, 1518-1525; *J. Nanosci. Nanotechnol.* 2005, 5, 1045-1049). Moreover, the changes in the shape of nanotubes affects their diffusion speed and hydrodynamic size (*ChemistrySelect* 2020, 5, 12570–12581; M. Instruments, Technical Note Malvern, MRK656-01 2012, 1). In fact, Perrier, Jolliffe, Scipioni, Geckeler, Fuchs and Scheinberg, et al. have also reported

similar bimodal size distributions in soft peptide-assembled nanotubes (*Adv. Mater.* 2013, 25, 1170–1172; *Macromol. Chem. Phys.* 2015, 216, 439–449), halloysite nanotubes (*Nanoscale* 2013, 5, 8577–8585), supramolecular nanotubes (*Angew. Chem. Int. Ed.* 2015, 54, 9376–9380), and even functionalized carbon nanotubes (*Nano Lett.* 2008, 8, 4221–4228), where the peak at larger diameters is related to the lengths of nanotubes or aggregation of particles, while the peaks at smaller hydrodynamic sizes is believed to be a rotational mode of nanotube motion and is inline with the anisotropic nature of the rodlike nanotubes (*Nano Lett.* 2008, 8, 4221–4228). According to the comment, to confirm the DLS results, we have passed P₄ sample through a 0.45 mm pore-sized syringe filter and **repeated DLS measurements for several times**. All the results presented bimodal size distributions. **We have added the new result** in Fig. 1b. Moreover, to further verify the tubular morphology of P₄, **we have repeated TEM and CLSM measurements and performed additional cryo-SEM analyses**. As seen from the images, the formation of tube-like morphology was evident, where cavities can be observed at the end of the nanotubes (Fig. 2i and Supplementary Fig. 11).

In addition, at higher TFA concentrations, the dense H-bonds that compact the vesicular membranes and shrink the particles were destabilized, thus P₃ exhibited a dramatic size increase and formed a swollen vesicular structure. For P₄ assembly, a higher concentration of TFA can break the H-bonding and n→π* interactions that dictates the secondary conformations, therefore, a nanotube-to-vesicle transition was observed. Interestingly, the particle sizes and morphologies could be restored to original ones after removal of TFA, and these processes could be repeated for many times. **We have performed additional experiments to confirm the reversibility of the transitions and provided the results** in Supplementary Fig. 17.

c. Comment: Figure 1h, the claim that the process can be implemented multiple times is not experimentally supported. The authors should add these characterizations.

Response: According to the thoughtful suggestion, **we have carried out additional DLS measurements** on P₁, P₂, and P₃ assemblies to demonstrate that the size change

process can be repeated multiple times. The results were provided in Supplementary Fig. 17 and analyzed in the revised manuscript (Page 9, Lines 7-8).

4) Molecular conformation.

a. Comment: The term beta sheet conformation is somehow not appropriate as evidenced in fig 2a (for instance, the positive peak at 200 nm is missing). It would be better to use sheet-like conformation or associate the conformational state to the creation of an intermolecular hydrogen bonding network.

Response: According to the helpful suggestion, we have used sheet-like conformation to describe the secondary structures of PUs in the revised manuscript (Page 1, Line 13; Page 11, Line 19; Page 12, Line 19; Page 13, Line 6; Page 13, Line 13; Page 16, Lines 8, 10 and 13; Page 20, Line 7; Page 21, Line 29).

b. Comment: Instead of using TFA, what happens if we add TFE or methanol? In both cases, the hydrogen bonds are strengthened with polypeptides (see Tim Deming's work for instance). On the other hand, is there an impact of the chirality of the amino acids (racemic mixture versus homochirality) on the conformation (or on the luminescent properties)?

Response: We appreciate the constructive comment. According to the suggestion, **we have added TFE and methanol to P₄ assemblies and carried out additional DLS, CD and TEM measurements.** The results were provided in Supplementary Fig. 24 and discussed in revised manuscript: *“Interestingly, the tube-to-vesicle transformation could also be found in the presence of structure-promoting solvents such as trifluoroethanol (TFE) and methanol (Supplementary Fig. 24) (Protein. Eng. 2000, 13, 739–743; J. Am. Chem. Soc. 2019, 141, 14530-14533). CD analysis confirmed that P₄ assemblies underwent a transition from sheet-like to helical structure after TFE or methanol addition, as evidenced by the appearance of two minima at 207 and 223 nm in the CD spectra (Supplementary Fig. 24). This may be because these solvents strengthened the intramolecular interactions and promoted the formation of helical*

conformations (Biochemistry 1992, 31, 8790-8798; Protein. Eng. 2000, 13, 739–743).
The exact mechanism requires further examination” (Page 13, Lines 10-18).

In addition, we also investigated the impact of the chirality of the amino acids on the conformation and luminescent properties of PUs. The results were provided in Fig.2a and Supplementary Figs. 20 and 27, and discussed in revised manuscript (Page 11, Lines 8-13; Page 15, Lines 16-20): “For comparison, we synthesized racemic PUs (DL-P₁ and DL-P₄) using LDI and D-Cys·OMe (Table 1, Supplementary Scheme 1, Figs. 18-19). CD characterization demonstrated that both samples adopted random coil structures in water (Supplementary Fig. 20). The results suggest that the conformation was influenced by the chirality and the number of segments.” (Page 11, Lines 8-13); “For comparison, the monomers and their mixture were nonfluorescent, and racemic DL-PUs showed much weaker fluorescence and lower quantum yields than corresponding PUs (Supplementary Figs. 26 and 27), indicating that both the formation of polyurea structures and chirality contributed to the interesting fluorescent property” (Page 15, Lines 16-20).

c. Comment: The following statement is wrong: “the results demonstrate for the first time a conformation-directed morphological transition in non-peptide polymers” (See for instance the work of Jing Sun on polypeptoids).

Response: To address the comment, we have revised the incorrect statement and changed it to: “Such a conformation-driven shape-shifting phenomenon in peptidomimetics is reminiscent of protein metamorphosis (ACS Chem. Biol. 2018, 13, 6, 1438–1446), which is helpful to understand the folding behaviors of biological macromolecules.” (Page 13, Lines 18-20)

5) Comment: Nonconventional fluorescence properties. The major point to review here is the scientific context. There is no real analogy with natural proteins from jellyfish. In the latter, the proteins have properties related to the presence of a fluorophore. The introduction of this paragraph should therefore be revised considering that this principle

of luminescence is already well documented and that key references are missing (see my comments in the introduction).

Response: Thanks for the suggestion. **We have revised the manuscript title, introduction and nonconventional fluorescence section to minimize the analogy with natural fluorescent proteins** (Page 1, Line 1; Page 1, Line 21; Page 2, Line 4; Page 3, Lines 9-10; Page 3, Line 12; Page 15, Lines 6-11). In addition, **some key references** regarding the luminescence phenomenon of nonconventional chromophore **have been cited** (references 52-59, 61) **and discussed** in the revised manuscript: “*Nonaromatic polyurethanes, polypeptides and other aliphatic compounds have been shown to generate noticeable intrinsic fluorescence under suitable conditions based on a clustering-triggered emission (CTE) mechanism (Polym. Chem. 2017, 8, 1722-1727; J. Polym. Sci., Part A: Polym. Chem. 2017, 55, 560-574; Mol. Syst. Des. Eng. 2018, 3, 364-375; Prog. Polym. Sci. 2019, 90, 35-117; Mater. Chem. Front. 2021, 5, 60-75; Chem. Soc. Rev. 2021, 50, 12616-12655). However, most of these polymers were fluorescent when concentrated or aggregated as powders and films but nonemissive in dilute solutions (Mol. Syst. Des. Eng. 2018, 3, 364-375; Small 2016, 12, 6586-6592). In this work, the compact H-bonding of PUs resulted in the close proximity of carbonyl moieties to form a “heterodox cluster”, which gave rise to emission under UV irradiation (Fig. 3a) (Macromolecules 2015, 48, 64-71). Of interest, the dilute PU solutions, assemblies and films emitted cyan blue and green fluorescence in the presence of UV irradiation” (Page 15, Lines 6-15); “It should be noted that Bonifacio et al. has observed pH-dependent weak fluorescence in polyurea dendrimers, and the light emission originated from the protonation of primary and tertiary amino groups (Angew. Chem. Int. Ed. 2012, 124, 5252-5255)” (Page 16, Lines 2-5).*

Minor comments:

- Comment: Table 1 from the esi should appear in the main manuscript (with correct SEC values).

Response: As suggested by the reviewer, we have placed Table 1 with corrected SEC

values in the revised main manuscript (Page 5, Line 6).

- Comment: In the intro., hydrophobic interactions do not exist (this is an entropic effect).

Response: The statement has been deleted in the revised manuscript (Page 2, Line 4).

- Comment: Ref 2-8 are not necessarily well chosen (see polypeptide polymers and secondary structures)

Response: We acknowledge the careful review. Now refs 2-8 have been carefully checked and more appropriate references regarding poly(amino acid)s and their secondary structures has been added (*Nat. Mater.* 2004, 3, 244-248; *Chem. Rev.* 2016, 116, 786-808; *Angew. Chem. Inter. Ed.* 2017, 56, 10826-10829; *J. Am. Chem. Soc.* 2018, 140, 6604-6610; *Chem. Soc. Rev.* 2018, 47, 7401-7425; *Adv. Mater.* 2020, 32, 2001108; *Angew. Chem. Inter. Ed.* 2021, 60, 22529-22536) (refs 4-10).

- Comment: Fig 1d, the label in water is confusing. This should be replaced by “dye alone”.

Response: We have replaced the label in water with “R6G”, and replaced other labels with “R6G@P₁, R6G@P₂, R6G@P₃ and R6G@P₄,” in Fig. 1d. Furthermore, we also added some explanation in the figure caption to make it clearer: “R6G dissolved in water and those encapsulated in PU assemblies (R6G@PUs) at the same concentration” (Page 6, Lines 7-8).

- Comment: Fig 4a: which PU is it?

Response: In Fig. 4a, PU represents the P₃ assemblies, which has been described in the figure caption (Page 17, Line 6).

To Reviewer #2

1. Comment: According to supplementary information, P5 was synthesized using LDI

and Lys•OMe•2HCl monomers. In Supplementary Table 1, however, P5 consists of only LDI residues. This is confusing and needs some explanation.

Response: We thank the reviewer for pointing out this mistake. As a control PU sample without disulfide bonds, P₅ was synthesized using L-Lysine ethyl ester diisocyanate (LDI) and L-lysine ethyl ester dihydrochloride (Lys.OEt.2HCl), therefore, P₅ should consist of L-lysine ethyl ester (Lys•OEt) residues rather than LDI residues. We have corrected this error in Table 1 (Page 5, Lines 6 and 8).

2. Comment: The size changes of all the polymer assemblies treated with trifluoroacetic acid (TFA) were presented in Figure 1e, but the change in size distributions of P4 was not shown in Figure 1f. Moreover, the volume fractions of TFA added were not specified.

Response: Now we have added the size distributions of P₄ in Fig. 1f. Moreover, the volume fraction of TFA added has been specified in the caption of Fig. 1f (Page 7, Line 1) and Fig. 2i (Page 10, Line 8).

3. Comment: Could the theory and characterization methods about the secondary conformations of poly (amino acid)s be applied to non-peptide polymers?

Response: Thanks for the constructive question. As we know, CD, FTIR and ThT binding assay have been established as powerful tools to characterize the secondary conformations of poly(amino acid)s (*Langmuir* 2005, 21, 4308–4315; *Macromolecules* 1969, 2, 624–628; *Nat. Commun.* 2011, 2, 206; *Nat. Commun.* 2017, 8, 92; *J. Am. Chem. Soc.* 2012, 134, 4112–4115; *J. Am. Chem. Soc.* 2019, 141, 14530–14533; *J. Am. Chem. Soc.* 2012, 134, 18542–18545; *Angew. Chem., Int. Ed.* 2013, 52, 9182–9186; *Proc. Natl. Acad. Sci. USA.* 2003, 100, 15463–15468). Interestingly, these techniques have also been widely used in studying the secondary structures of non-peptide polymers, despite that some of the polymers did not have chirality. For instance, oligomeric *N*-substituted glycines (or “peptoids”) can adopt stable helices which exhibited intense CD spectra that closely resemble those of peptide α -helices (*J. Am. Chem. Soc.* 2001, 123, 2958–

2963; *J. Am. Chem. Soc.* 2001, 123, 6778–6784). Yang et al. proved that L and D-tyrosine-derived polyurethanes showed right- and left-handed helical conformation using CD spectroscopic and FTIR analysis (*Polymer* 2011, 52, 3745-3751). Pappas and Otto et al. demonstrated the formation of β -sheet type assembly in a fully synthetic and structurally complex foldamer using CD spectra and ThT experiment (*J. Am. Chem. Soc.* 2021, 143, 7388–7393). In our work, the PUs were constructed from natural amino acids and showed abundant H-bond donors and acceptors in the polymeric backbone and side chains, hence, they can form interesting helical and sheet-like conformations, which were well characterized with CD, FTIR, ThT techniques. In our future work, we would like to explore the potential of these methods in other non-peptide polymeric systems.

4. Comment: The fluorescence behaviors of the polymers were fascinating. What about the fluorescence quantum yield of the polymers?

Response: According to the comment, **we have determined the fluorescence quantum yield** of PUs in reference to quinine sulfate. **The experimental details have been specified** in the revised Supplementary information (Page 4, Lines 10-21). The results were presented in Supplementary Fig. 26 and discussed in the revised manuscript: “*Of interest, the dilute PU solutions, assemblies and films emitted cyan blue and green fluorescence in the presence of UV irradiation (Fig. 3b and Supplementary Fig. 25), with high fluorescence quantum yields up to 29% (Supplementary Fig. 26). For comparison, the monomers and their mixture were nonfluorescent, and racemic DL-PUs showed much weaker fluorescence and lower quantum yields than corresponding PUs (Supplementary Fig. 27), indicating that both the formation of polyurea structures and chirality contributed to the interesting fluorescent property.*” (Page 15, Lines 13-20).

5. Comment: In the animal study, the change of the body weights of the mice should be provided.

Response: The change of the body weights of the mice has been provided in Supplementary Fig. 37 and analyzed in the revised manuscript: “*The mice injected with DOX@PUs did not show apparent loss of body weight during the days of injection compared with those treated with free DOX (Supplementary Fig. 37), implying reduced systemic toxicity of polymeric formulations*” (Page 20, Line 22; Page 21, Lines 1-3)

6. Comment: In addition, the scale bars should be added in the confocal laser scanning microscope (CLSM) images.

Response: We have added the scale bars in the CLSM images (Fig. 1c, 1g, Supplementary Fig. 23 and 33).

7. Comment: There are a couple of minor grammar or type errors in the manuscript, for instance “Th T” should be “ThT” in Figure 2b.

Response: As suggested, “Th T” has been replaced with “ThT” in Figure 2b. Moreover, we have carefully checked the whole manuscript and supplementary document for grammar problems and type errors, and corrected them appropriately.

8. Comment: Some typos in the reference list should be corrected, and the recently published review or research articles should be discussed in the revision, for example, 10.15212/bioi-2021-0016, 10.1016/j.nantod.2019.100800.

Response: We have carefully checked the reference list and corrected the formatting errors. In addition, the recommended references (*Nano Today* 2019, 29, 100800; *BIO Integr.* 2021, 2, 57-60) have been cited in the revision (refs 66 and 68).

Overall, we have carried out additional experiments, responded in detail to each point of the reviewer, and revised the manuscript according to the comments. Moreover, we have carefully checked the manuscript and supporting document for grammar problems and other mistakes, and revised them appropriately. In addition, Ms. Jinling Zhao has made significant contribution in performing additional experiments and revising the

manuscript, hence, her name has been added in the author list. **All changes made are in red color as shown** in the documents “**Revised manuscript**” and “**Highlighted revised supporting information**”. We hope, with these modifications and improvements, the new manuscript would be suitable for publication in *Nature communications*.

The change of manuscript and supporting information **were also listed as follows** (unless otherwise noted, all locations shown below are directed to the original manuscript):

1. Changes in Manuscript

- (1) Page 1, Line 1, “Fluorescent protein-mimic” was changed to “Bioinspired Fluorescent”.
- (2) Page 1, Line 3, “Jinling Zhao¹,” was added.
- (3) Page 1, Line 7, “or dmmshx@163.com.” was deleted.
- (4) Page 1, Line 8, “Pseudo-polypeptides” was changed to “Pseudo-poly(amino acid)s”.
- (5) Page 1, Line 11, “protein-mimic” was changed to “protein-inspired”.
- (6) Page 1, Line 12, “PUs” was deleted.
- (7) Page 1, Line 13, “weight-dependent β -sheet-like conformations and” was changed to “weight- and solvent-dependent helical and sheet-like conformations as well as”.
- (8) Page 1, Line 13, “protein-inspired” was changed to “protein-mimic”.

- (9) Page 1, Line 14, “aggregate” was changed to “aggregation”; “(AIE)” was deleted.
- (10) Page 1, Line 14, “character” was changed to “characteristics”.
- (11) Page 1, Line 21, “protein-mimic” was changed to “bioresponsive”.
- (12) Page 2, Lines 3-7, “In nature, proteins can fold into unique three-dimensional structures through hydrophobic interactions, hydrogen-bonding (H-bonding) and electrostatic interactions, which play crucial roles in different structural and biological functions, such as metamorphosis, sensing, and transportation.¹ To understand and imitate the structure and function of proteins, natural amino acid-based polymers such as poly(amino acid)s” was changed to “Natural amino acid-derived polymers are recognized as promising materials to understand the structures and functions of proteins and develop bioinspired functional biomaterials for use in biomimicry, drug delivery and tissue engineering applications.¹⁻³ Among them, poly(amino acid)s (PAAs)”.
- (13) Page 2, Line 9, “ordered” was changed to “specific”.
- (14) Page 2, Line 9, “including” was changed to “such as”.
- (15) Page 2, Line 10, “a good variety of pseudo-polypeptides that do not contain peptide bonds” was changed to “a family of pseudo-PAAs”.
- (16) Page 2, Line 12, “anhydrite, triazole” was deleted; “ α -amino acids into the polymeric backbones via” was added.
- (17) Page 2, Line 13, “moieties into the polymeric backbones” was changed to “linkages”.
- (18) Page 2, Line 14, “higher molecular tunability” was changed to “highly tunable molecular architecture”.
- (19) Page 2, Line 15, “Unfortunately” was changed to “However”.
- (20) Page 2, Line 16, “linkages” was changed to “moieties”.
- (21) Page 2, Line 17, “ordered” was changed to “order hierarchical”; “biological” was added.
- (22) Page 2, Line 18, “H-bonding” was changed to “Hydrogen-bonding (H-bonding)”.
- (23) Page 2, Line 21, “hydrogen bonding” was changed to “H-bonding”.

- (24) Page 3, Line 2, “hydrogen bonding” was changed to “H-bonding”.
- (25) Page 3, Line 4, “It is thus hypothesized that the incorporation of urea motifs into pseudo-poly(amino acid)s may impart abundant H-bonds and, in principle, can assist the folding of polymers into various hierarchical structures. However, to our knowledge, polyureas (PUs) with ordered conformations and protein-like functions has not been reported so far” was changed to “In fact, sequence-defined oligoureas have been established as attractive peptidomimetic foldamers that adopt robust helical conformation stabilized by three-center H-bonds.¹⁹⁻²¹ As a class of urea-rich polymers, polyureas (PUs) have been widely used as elastomers, coatings, adhesives and biomaterials.²²⁻²⁴ Nonetheless, PU-based functional pseudo-PAAAs with ordered secondary structures and conformation-mediated functionalities have been rarely reported.”.
- (26) Page 3, Line 9, “reported” was changed to “report”.
- (27) Page 3, Line 9, “a protein-mimic” was changed to “protein-inspired”.
- (28) Page 3, Line 10, “showed” was changed to “show”.
- (29) Page 3, Line 10, “weight-dependent conformations” was changed to “weight- and solvent-dependent conformations”.
- (30) Page 3, Line 11, “green fluorescent protein-inspired fluorescence with” was deleted.
- (31) Page 3, Line 12, “character” was changed to “properties”;
- (32) Page 3, Line 13 “exhibited” was changed to “exhibit”.
- (33) Page 3, Line 19, “Lysine” was changed to “lysine”.
- (34) Page 3, Lines 19-21, “L-” was added.
- (35) Page 3, Line 20, “ethylester” was changed to “ethyl ester”.
- (36) Page 4, Line 3, “DLS” was changed to “dynamic light scattering (DLS)”.
- (37) Page 4, Line 4 “in TEM images” was added; “and full TEM images are shown in Supplementary Fig. 10” was added.
- (38) Page 4, Line 5 “The scale bars are 100 nm” was added.
- (39) Page 4, Line 6 “dissolved” was added; “those encapsulated in” was added.

- (40) Page 4, Line 7 “(R6G@PUs)” was added.
- (41) Page 4, Line 8“(16 v%)” was added; “with TFA treatment” was changed to “treated with TFA.
- (42) Page 4, Line 10, “The scale bars are 0.5 and 2 μ m for TEM and CLSM, respectively.” was added.
- (43) Page 5, Line 5, “confirm” was changed to “confirmed”.
- (44) Page 5, Line 7, “LDI” was changed to “Lys·OEt”.
- (45) Page 5, Line 10, “Supplementary Table 1” was changed to “Table 1”.
- (46) Page 5, Line 12, “To verify the triblock architecture of PUs, we synthesized a control polymer containing diblock and triblock mixtures (DTPU) by feeding half amount of mPEG. As expected, DTPU exhibited much larger polydispersity (\bar{M}_w/\bar{M}_n 1.92), bimodal molecular weight distribution (Supplementary Fig. 7), and reduced integral area of PEG in the ^1H NMR spectrum (Supplementary Fig. 8). It is believed that diblock or non-PEGylated PUs possess one or two amine groups on the chain end due to the reaction of isocyanate-terminated prepolymers with water.²⁵ Therefore, we carried out an end-group analysis taking P₃ and DTPU as examples. The polymers were reacted with FITC and subjected to UV-vis and fluorescence measurements. Apparently, DTPU displayed distinct UV absorption at 490 nm and a fluorescence emission peak at 552 nm (Supplementary Fig. 9), suggesting that it has been successfully labeled with FITC. In contrast, P₃ did not show noticeable FITC signal (Supplementary Fig. 9), thus confirming the PEGylated triblock structure of PUs that do not have any reactive groups.” was added.
- (47) Page 5, Line 12, Table 1 and its caption were added.
- (48) Page 5, Line 20, “Supplementary Fig. 10” was added.
- (49) Page 5, Line 20, “Further cryo-scanning electron microscopy (cryo-SEM) image of P₄ presents a number of open-ended tubular structures with large aspect ratios (Supplementary Fig. 11).” was added.
- (50) Page 5, Line 21, “and doxorubicin hydrochloride (DOX·HCl)” was added.

- (51) Page 5, Line 22, “a hydrophilic probe” was changed to “hydrophilic probes”; “dyes” was changed to “R6G”.
- (52) Page 6, Line 1, “R6G” was changed to “the dye”.
- (53) Page 6, Line 2, “Moreover, the DOX fluorescence was detected in both emission spectra (Supplementary Fig. 12) and confocal laser scanning microscopy (CLSM) images with typical spherical and tubular architectures (Fig. 1c), which agrees well with TEM observation.” was added.
- (54) Page 6, Line 2, “This result reveals” was changed to “These results reveal”; “assembly is” was changed to “assemblies are”.
- (55) Page 6, Line 3, “pocket” was changed to “pockets”; “vesicles” was changed to “nanovessels”.
- (56) Page 6, Line 5, “Supplementary Fig.7” was changed to “Supplementary Fig.13”.
- (57) Page 6, Line 6, “To provide more direct evidence for the vesicular and nanotube structures, we encapsulated a hydrophilic doxorubicin hydrochloride (DOX·HCl) within the PU dispersions. The DOX fluorescence was detected in both emission spectra (Supplementary Fig. 8) and confocal laser scanning microscopy (CLSM) images with typical spherical and tubal architectures (Fig. 1c), which agrees well with TEM observation.” was deleted.
- (58) Page 6, Line 18, “Note that the TFA treatment did not the disrupt the molecular integrity of PUs (Supplementary Fig. 14).” was added.
- (59) Page 7, Line 1, “confirm” was changed to “confirmed”.
- (60) Page 7, Line 5, “allows” was changed to “allowed”.
- (61) Page 7, Line 6, “Supplementary Figs. 10-11” was changed to “Supplementary Fig. 16”.
- (62) Page 7, Line 9, “Fig. 1f” was changed to “Fig. 1 e and f”.
- (63) Page 7, Line 10, “plays” was changed to “played”.
- (64) Page 7, Line 12, “Supplementary Fig. 17” was added; “for” was deleted.
- (65) Page 7, Line 18, “as chiral tyrosine-constructed polyurethanes have shown helical configurations” was added.

- (66) Page 8, Line 8, “CLSM (left) and cryo-SEM (right) images of P₄ assemblies treated with 16% TFA. The scale bar is 1 μm” was changed to “cryo-SEM images of P₄ assemblies before and after TFA treatment (16 v%). The left and right scale bars are 100 nm and 1 μm, respectively”.
- (67) Page 8, Line 11, “For comparison, we synthesized racemic PUs (DL-P₁ and DL-P₄) using LDI and D-Cys·OMe (Table 1, Supplementary Scheme 1, Figs. 18-19). CD characterization demonstrated that both samples adopted random coil structures in water (Supplementary Fig. 20). The results suggest that the conformation was influenced by the chirality and the number of segments.” was added.
- (68) Page 9, Line 1, “Hence, it has” was changed to “and has”.
- (69) Page 9, Line 9, “(Fig. 2c),” was deleted.
- (70) Page 9, Line 14, “(Supplementary Fig. 12 and Supplementary Table 2)” was changed to “(Supplementary Fig. 21 and Supplementary Table 1)”.
- (71) Page 9, Line 21, “shows” was changed to “showed”.
- (72) Page 10, Line 8, “an interplanar space” was changed to “distances”.
- (73) Page 10, Line 9, “random coil structures (Supplementary Fig. 13)” was changed to “a random coil structures (Supplementary Fig. 22)”.
- (74) Page 10, Line 13, “of peptides” was added.
- (75) Page 10, Line 15, “β-sheet-like conformation” was changed to “sheet-like conformation in PUs”.
- (76) Page 10, Line 18, “dynamic light scattering (DLS), cryo-scanning electron microscopy (cryo-SEM), TEM and CLSM experiments (Fig. 2i-g, Supplementary Fig. 14).” was changed to “DLS, cryo-SEM, TEM and CLSM experiments (Figs. 2i-g, Supplementary Fig. 23).”.
- (77) Page 10, Line 19, “The results demonstrate for the first time a conformation-directed morphological transition in non-peptide polymers.” was changed to “Interestingly, the tube-to-vesicle transformation could also be found in the presence of structure-promoting solvents such as trifluoroethanol (TFE) and

methanol (Supplementary Fig. 24).^{49, 50} CD analysis confirmed that P₄ assemblies underwent a transition from sheet-like to helical structure after TFE or methanol addition, as evidenced by the appearance of two minima at 207 and 223 nm in the CD spectra (Supplementary Fig. 24). This may be because these solvents strengthened the intramolecular interactions and promoted the formation of helical conformations.^{49, 51} The exact mechanism requires further examination. Such a conformation-driven shape-shifting phenomenon in peptidomimetics is reminiscent of protein metamorphosis,⁵² which is helpful to understand the folding behaviors of biological macromolecules”.

- (78) Page 11, Line 4, “1 mg/mL” was changed to “1 mg mL⁻¹”; “Lys·OEt” was changed to “L-Lys·OEt” ; “by dialysis” was deleted.
- (79) Page 12, Line 3, “Nonaromatic polyurethanes, polypeptides and other aliphatic compounds have been shown to generate noticeable intrinsic fluorescence under suitable conditions based on a clustering-triggered emission (CTE) mechanism.⁵³⁻⁵⁸ However, most of these polymers were fluorescent when concentrated or aggregated as powders and films but nonemissive in dilute solutions.^{55, 59}” was added.
- (80) Page 12, Line 3, “In nature, the folding of peptide chains in jellyfish fluorescent protein results in the “close contact” of amino acid residues for bioreaction to form fluorescent chromophore.⁴⁴⁻⁴⁶” was deleted.
- (81) Page 12, Line 7, “gives” was changed to “gave”; “in the presence of” was changed to “under”.
- (82) Page 12, Line 8, “polymer” was changed to “dilute PU”.
- (83) Page 12, Line 9, “under UV irradiation (Fig. 3b and Supplementary Fig. 15)” was changed to “in the presence of UV irradiation (Fig. 3b and Supplementary Fig. 25), with high fluorescence quantum yields up to 29% (Supplementary Fig. 26).”;
- (84) Page 12, Line 10, “(Fig. 3b)” was deleted.
- (85) Page 12, Line 11, “and racemic DL-PU_s showed much weaker fluorescence and lower quantum yields than corresponding PU_s (Supplementary Fig. 27),” was

- added.
- (86) Page 12, Line 11, “the formation of polyurea structures are the major contributions” was changed to “both the formation of polyurea structures and chirality contributed”.
- (87) Page 12, Line 13, “green” was changed to “blue”.
- (88) Page 12, Line 14, “(GFP) (λ_{ex} 395 nm, λ_{em} 475 nm)” was changed to “(BFP) (λ_{ex} 383 nm, λ_{em} 448 nm)”.
- (89) Page 12, Line 15, “character” was changed to “characteristics”; “increased” was changed to “increasing”.
- (90) Page 12, Line 19, “is” was changed to “was”.
- (91) Page 12, Line 20, “which is rarely reported in linear aliphatic 21 polymers.” was deleted.
- (92) Page 13, Line 2, “distributions of PU” was changed to “distribution of P₃”.
- (93) Page 14, Line 4, “While the conformation changed from sheet-like to random coil structure (Supplementary Fig. 28),” was added.
- (94) Page 14, Line 9, “accompanied by the” was changed to “with”.
- (95) Page 14, Line 10, “protein aequorin” was changed to “proteins” .
- (96) Page 14, Line 11, “emission” was changed to “emissions”; “green fluorescence proteins” was added.
- (97) Page 14, Line 15, “(Fig. 3g-h)” was changed to “(Figs. 3g-h)”.
- (98) Page 14, Line 22, “methoxy poly(ethylene glycol)-poly(ϵ -caprolactone)” was added.
- (99) Page 15, Line 1, “(Supplementary Fig. 17)” was changed to “(Supplementary Fig. 30)”.
- (100) Page 15, Line 9, “(DAPI, blue)” was changed to “DAPI (blue)”.
- (101) Page 16, Line 3, “completed” was changed to “complete”.
- (102) Page 16, Line 7, “(Fig. 4c-d, Supplementary Fig. 18).” was changed to “(Figs. 4c-d, Supplementary Fig. 31)”.
- (103) Page 16, Line 11, “in the presence of 10 mM GSH” was changed to “after GSH

- treatment (10 mM)".
- (104) Page 16, Line 12, "(Supplementary Fig. 19)." was changed to "(Supplementary Fig. 32)".
- (105) Page 16, Line 15, "pseudo-poly(amino acid)" was changed to "pseudo-PAA".
- (106) Page 16, Line 19, "(Fig. 5a-c, Supplementary Fig. 20)." was changed to "(Figs. 5a-c, Supplementary Fig. 33)".
- (107) Page 16, Line 20, "nano-tubal" was changed to "nano-tubular".
- (108) Page 16, Line 22, "(Fig. 5d-g)" was changed to "(Figs. 5d-g)".
- (109) Page 17, Lines 4-5, "µg/mL" was changed to "µg mL⁻¹".
- (110) Page 17, Lines 5-7, "(Supplementary Fig. 23, Supplementary Table 3).", "(Supplementary Fig. 22)." was changed to "(Supplementary Fig. 34, Supplementary Table 2).", "(Supplementary Fig. 35)".
- (111) Page 17, Line 13, "The mice injected with DOX@PUs did not show apparent loss of body weight during the days of injection compared with those treated with free DOX (Supplementary Fig. 37), implying reduced systemic toxicity of polymeric formulations" was added.
- (112) Page 17, Line 13, "found" was deleted.
- (113) Page 18, Line 4, "DOX loaded" was changed to "DOX-loaded".
- (114) Page 18, Line 8, "5.0 mg/kg" was changed to "5.0 mg kg⁻¹".
- (115) Page 19, Line 5, "pseudo-polypeptides" was changed to "bioresponsive polymers".
- (116) Page 19, Line 10, "pseudo-poly(amino acid)" was changed to "pseudo-PAA".
- (117) Page 19, Line 11, "emission" was changed to "emissions".
- (118) Page 19, Line 12, "interesting conformational transitions" was added.
- (119) Page 20, Line 2, "Cys·OMe·2HCl" was changed to "L-Cys·OMe·2HCl"; "D-Cys·OMe·2HCl" was added.
- (120) Page 20, Line 2, "Lys·OEt·2HCl" was changed to "L-Lys·OEt·2HCl".
- (121) Page 20, Line 6, "5.5" was added.
- (122) Page 20, Line 9, "(80-90 % yield)" was added.

- (123) Page 20, Line 30, “Dulbecco’s Modified Eagle’s Medium” was added.
- (124) Page 21, Line 10, “treated with DOX@MPU and DOX@CMPU” was changed to “administrated with DOX@P₅, DOX@P₂ and DOX@P₄”.
- (125) Page 21, Line 17, “hematoxylin-eosin (H&E) staining, terminal deoxynucleotidyl transferase-mediated deoxyuridine triphosphate nick end labeling (TUNEL) assay and nuclear-associated antigen (Ki67)” was changed to “H&E staining, TUNEL assay and Ki67”.
- (126) Page 21, Line 19, “Data availability” was added
- (127) Page 21, Line 20, “The data that support the findings of this study are available within the paper and its Supplementary Information files and available from the corresponding author upon request. Source data are provided with this paper.” was added.
- (128) Page 24, Line 20, “J.Z” was added.

In addition, Figs 1a, 1d, 1g and Fig.2b were modified; Figs 1b,1c and 1f were updated; Fig.2i was replaced with cryo-TEM image with higher quality; Supplementary Table 1 and caption were moved to revised manuscript (Table 1); New references were added and the reference list was updated.

2. Changes in Supporting Information

- (1) Page 1, Line 3, “Fluorescent protein-mimic polyureas toward conformation-assisted metamorphosis, discoloration and intracellular drug delivery” was changed to “Bioinspired Fluorescent Polyureas toward Conformation-Assisted Metamorphosis, Discoloration and Intracellular Drug Delivery”.
- (2) Page 1, Line 6, “Jinling Zhao¹,” was added.
- (3) Page 1, Line 10, “*dmmshx@163.com*” was deleted.
- (4) Page 2, Line 2, “Cystine dimethyl ester dihydrochloride (Cys·OMe·2HCl, 98%) was” was changed to “L-Cystine dimethyl ester dihydrochloride (L-Cys·OMe·2HCl, 98%) and D-cystine dimethyl ester dihydrochloride (D-

Cys·OMe·2HCl, 98%) were”.

- (5) Page 2, Line 4, “was” was changed to “were”.
- (6) Page 2, Line 6, “Lys·OEt·2HCl” was changed to “L-Lys·OEt·2HCl”.
- (7) Page 2, Line 9, “trifluoroethanol (TFE)” was added; “Sodium” was changed to “sodium”.
- (8) Page 2, Line 14, “Dimethylsulfoxide-D6” was changed to “Perdeuterated dimethylsulfoxide”; “Thioflavin T” was changed to “thioflavin T”.
- (9) Page 2, Line 36, “at room temperature using THF as an eluent.” was changed to “at 40 °C using DMF/LiBr (2 g L⁻¹) as an eluent and polymethyl methacrylate (PMMA) as a standard.”.
- (10) Page 2, Line 37, “and polymethyl methacrylate (PMMA) as a standard.” was added.
- (11) Page 2, Line 39, “Cryo” was changed to “cryo”.
- (12) Page 2, Line 40, “samples” was changed to “sample”.
- (13) Page 2, Line 42, “Cryo” was changed to “cryo”.
- (14) Page 3, Line 34, “3.06 ppm (CH-CH₂-S-S-CH₂-CH) was added.
- (15) Page 3, Line 35 “at 2.96 (NH-CH₂-CH₂-CH₂-CH₂)” was added.
- (16) Page 3, Line 36 1.94 (NH-CH₂-CH₂-CH₂-CH₂) ppm” was added.
- (17) Page 3, Line 40, “LDI” was added ; “Cys·OMe” was added.
- (18) Page 4, Line 7, “Fluorescence Quantum yield (FQY) measurement. The FQYs of PUs were measured in reference to quinine sulfate in 0.1 M H₂SO₄ (literature quantum yield 54% at 350 nm excitation). The same excitation wavelength and slit band widths were applied for the two samples. The formula used for FQY measurements was as follows.

$$\text{FQY} = (I / I_{\text{R}}) \times (A_{\text{R}} / A) \times (\eta / \eta_{\text{R}})^2 \times \text{FQY}_{\text{R}} \quad (1)$$

where FQY is the quantum yield of the sample, I is the integral area under the fluorescence spectrum, η is the refractive index of the solvent used and A is the absorbance at the excitation wavelength. The subscript R represents the reference. To minimize reabsorption effects, absorbencies were kept under 0.05

at the excitation wavelength of 350 nm. the FQYs of PUs and DL-PUs were measured and presented in Supplementary Fig. 26. The FQY of PUs is up to 29%, which is much higher than those of the reported polysiloxanes, poly(amino esters) and poly(urea-urethane)s.⁵⁻⁸” was added.

- (19) Page 4, Lines 33, “equation (1).⁷⁻⁹” was changed to “equation (2).^{9, 10}”.
- (20) Page 4, Lines 34, “(1)” was changed to “(2)”.
- (21) Page 4, Lines 35, “were” was changed to “are”.
- (22) Page 5, Lines 6, “and size” was added.
- (23) Page 5, Lines 7, “DLS” was added.
- (24) Page 5, Lines 9, “and P₃” was changed to “P₃, and P₄”.
- (25) Page 5, Lines 16, “The size and size distribution of PUs” was changed to “To investigate the reversibility of morphological transition, the sizes and size distributions of PU”.
- (26) Page 5, Lines 18, “. The” was changed to “, and the”.
- (27) Page 5, Lines 18, “PUs assemblies treated with TFA” was changed to “the samples”.
- (28) Page 5, Line 19, “a λ_{ex} ” was changed to “ λ_{ex} ”.
- (29) Page 5, Lines 19, “After removal of TFA by extensive dialysis (MWCO 3500), the solution was adjusted to a constant volume and determined again with DLS and fluorescence spectrophotometer. The removal and addition of TFA were repeated many times for the measurements. In addition, to verify the integrity of PU structures during TFA treatments, the polymers were incubated in TFA for one week and analyzed with ¹H NMR. As found in Supplementary Fig. 14, the characteristic peaks and their integral areas were kept nearly unchanged after incubation with TFA, suggesting that the TFA treatment did not disrupt the molecular integrity of PUs.” was added.
- (30) Page 5, Line 20, “Pus” was changed to “PU assemblies”.
- (31) Page 5, Line 20, “taking P₃ as an example and a diblock copolymer of mPEG and poly(ϵ -caprolactone) (mPEG-PCL) as a control” was added.

- (32) Page 5, Line 21, “PU assemblies” was changed to “polymeric assemblies”;
- (33) Page 5, Line 21, “was added with methanol and incubated with sodium dodecyl sulfate (SDS, 0.02 M) with shaking.” was changed to “or treated with methanol or SDS (0.02 M) with shaking.” .
- (34) Page 5, Line 22, “Then the” was added.
- (35) Page 5, Line 25, “MTT assay was performed to” was changed to “To”.
- (36) Page 5, Line 25, “empty PU assemblies and drug-loaded formulations, using DOX as a model drug. Briefly, MCF-7 cells” was changed to “drug-free and drug-loaded PU assemblies, MCF-7 cells”.
- (37) Page 5, Line 27, “Then the free drugs, and drug-loaded PUs formulations with different drug concentration were separately added” was changed to “Then the samples with different concentrations were added separately”.
- (38) Page 5, Line 37, “MCF-7 cells were large-scale expanded ex vivo in culture medium and collected in PBS.” was changed to “MCF-7 cell line was cultured in Dulbecco’s Minimal Eagle Medium (DMEM) medium with 10% fetal bovine serum”,
- (39) Page 5, Line 38, “the right armpit of BABL/c nude mice” was changed to “drug-free and drug-loaded”, “the upper right flank of the mice.” .
- (40) Page 5, Line 39, “weight” was changed to “weights”; “size” was changed to “sizes”.
- (41) Page 5, Line 43, “PUs” was changed to “PU”.
- (42) Page 6, Line 14, “Synthesis of PUs (P₁, P₂, P₃, P₄). reagents and conditions: (a) TEA, 60 ° C, 2 h; (b) mPEG 5000, stannous octanoate, 80 ° C, 3 d.” was changed to “Synthesis of PUs and DL-PUs (P₁, P₂, P₃, P₄, DL-P₁, DL-P₄) using L-Cys·OMe·2HCl, D-Cys·OMe·2HCl, L-Lys·OEt·2HCl and LDI as monomers and mPEG5000 as an end-capping agent. Reagents and conditions: (a) TEA, DMAC, 60 °C, 2 h; (b) mPEG5000, stannous octanoate, 80 °C, 3 d, dialysis, lyophilization (80-90% yield).”.
- (43) Page 6, Line 14, “synthesis of P₅. Reagents and conditions: (a) TEA, 60 ° C, 2 h;

- (b) mPEG 5000, stannous octanoate, 80 ° C, 3 d.” was changed to “Synthesis of P₅ using L-Lys·OEt·2HCl and LDI as monomers and mPEG5000 as an end-capping agent. Reagents and conditions: (a) TEA, DMAC, 60 °C, 2 h; (b) mPEG5000, stannous octanoate, 80 °C, 3 d, dialysis, lyophilization (90% yield).”.
- (44) Page 7, Line 2, “(a) P₁; (b) P₂; (c) P₃; (d) P₄” was added.
- (45) Page 8, Line 2, “in DMSO-*d*₆” was added.
- (46) Page 9, Line 2, “in DMSO-*d*₆: (a) P₁; (b) P₂; (c) P₃; (d) P₄” was added.
- (47) Page 10, Line 2, “in DMSO-*d*₆” was added.
- (48) Page 12, Line 2, “diagrams” was changed to “chromatograms”; “in DMF/LiBr (2 g L⁻¹).” was added.
- (49) Page 13, Line 3, “emission (a)” was changed to “emission spectra ($\lambda_{\text{ex}} = 334$ nm)” ; “excitation spectra (b, $\lambda_{\text{em}} = 372$ nm),” was changed to “(b) excitation spectra ($\lambda_{\text{em}} = 372$ nm)” ; “PUs” was changed to “PU assemblies”; “ $\lambda_{\text{ex}} = 334$ nm” was added; “spectra” was added.
- (50) Page 14, Line 2, “FL” was changed to “Fluorescence”; “DOX-encapsulated” was added; “DOX@PUs” was added.
- (51) Page 15, Line 2, “PUs with different volume of TFA (n = 0, 4, 10, 16): a, P₁; b, P₂; c, P₃; d, P₄.” was changed to “PU assemblies in aqueous solutions containing different concentrations of TFA (0, 4, 10, 16 v%): (a) P₁; (b) P₂; (c) P₃; (d) P₄.”.
- (52) Page 15, Line 2, “micrograph” was changed to “micrographs”; “PUs loaded FITC and DOX” was changed to “FITC- and DOX-coloaded PU assemblies”; “The scale bar is 4 μ m” was added.
- (53) Page 19, Line 3, “Conformation” was changed to “Structure”.
- (54) Page 20, Line 4, “Typical TEM image of P₄ assemblies with TFA treatment. the scale bar is 500 nm.” was changed to “CLSM (left) and TEM (right) images of P₄ assemblies treated with TFA (16 v%). The left and right scale bars are 2 μ m and 1 μ m, respectively.”.
- (55) Page 21, Line 3, “spectra” was changed to “spectrum”; “solutions” was changed to “solution”; “in DMF (0.5 mg mL⁻¹) ($\lambda_{\text{ex}} = 365$ nm)” was added.

- (56) Page 22, Line 4, “PU assemblies (a) and PEG-PCL (b) diluted” was changed to “P₃ assemblies (a) and mPEG-PCL assemblies (b) before (a) and after dilution”.
- (57) Page 23, Line 3, “PUs assemblies (a) and PEG-PCL (b) with 4 different substance treatment.” was changed to “P₃ assemblies (a) and mPEG-PCL assemblies (b) before and after treatments with methanol or SDS.” .
- (58) Page 24, Line 2, “free R6G dissolved in water and those” was added; “in the absence of GSH” was added; “with 0 mM GSH” was changed to “assemblies incubated” .
- (59) Page 25, Line 2, “QD-loaded PU assemblies incubated with 10 mM of GSH and 0 mM GSH for different times. P₃ (a), P₅ (b),” was changed to “QD-loaded P₃ assemblies incubated with 0 mM of GSH (a) and those of P₅ assemblies incubated with 10 mM of GSH (b) for different times. (c) Normalized decrease in” .
- (60) Page 26, Line 3, “The scale bars are 5 μm” was added.
- (61) Page 30, Line 9, “hydrogen-bonded” was changed to “H-bonded”.
- (62) Page 30, Line 14, “μg/mL” was changed to “μg mL⁻¹”.

In addition, new figures (Supplementary Figs 7-11, 14, 17-20, 24, 26-28, 37) were added; Supplementary Schemes 1-2 were modified; Supplementary Figs 1-4, 6 were modified; The numbers of Supplementary Figs were updated. Moreover, Supplementary Table 1 and caption were moved to revised manuscript; new supporting references were added and the reference list was updated.

REVIEWER COMMENTS

Reviewer #1 (Remarks to the Author):

The work of Mingming Ding and coworkers reports an interesting approach towards macromolecular peptidomimetics (and not pseudo-PAA) with non-conventional luminescent properties. This revised manuscript do not fully address all the concerns raised in the previous revision. Some significant questions have not been answered in this revision. I therefore cannot yet recommend the publication of this paper in Nature Communications if the previous comments are not all addressed.

1) Title. The term “bioinspired” is not appropriate and should be replaced by “intrinsically fluorescent “: Intrinsically Fluorescent Polyureas toward Conformation-Assisted Metamorphosis, Discoloration and Intracellular Drug Delivery.

2) Bibliography: The terms referring to peptides or aminoacids are still not appropriate and should be replaced (pseudo-poly(amino acid)s, etc.). The term peptidomimetic is correct but I accept that it may be too broad. The reader should not be misled here. The article is not really dealing with amino acids (C monomer unit) and the polymer is PU. If the term peptidomimetic is not used, I suggest using the term PU-based pseudo peptide to simplify/clarify the concept (for instance, p 3, lines 7-9). Finally, the references given in during the first revision should all be taken into account.

3) Figure 1c. Why not using intrinsic fluorescence instead? This would be more convincing given the results developed by the authors.

4) The quality of TEM images is still poor and the presence of nanotubes is still not completely verifiable. The authors should surely nuance this interpretation by indicating that it is a hypothesis that remains to be verified (SAXS etc.). Nanofiber formation is certainly the correct interpretation and the DLS measurement issues have not been properly accounted for.

5) Esi. Figure 20: The units are missing in the analysis presented (molar ellipticity). As it is, it is difficult to conclude on the basis of the data unless the non-racemized polymer is overlaid to compare.

6) Esi. Please provide the integrations on the NMR analyses (fig. 8 for instance but also for the others).

Reviewer #2 (Remarks to the Author):

The revision is ready for publication.

Response to the comments on NCOMMS-21-36124A

To Reviewer #1

Comment: The work of Mingming Ding and coworkers reports an interesting approach towards macromolecular peptidomimetics (and not pseudo-PAA) with non-conventional luminescent properties. This revised manuscript do not fully address all the concerns raised in the previous revision. Some significant questions have not been answered in this revision. I therefore cannot yet recommend the publication of this paper in Nature Communications if the previous comments are not all addressed.

Response: We are grateful to the reviewer's support for our work and the kind reminding. Now we have fully addressed all the concerns raised in both the previous revision and the new comments. We hope, with these modifications and improvements, the new manuscript would be suitable for publication.

1) Comment: Title. The term "bioinspired" is not appropriate and should be replaced by "intrinsically fluorescent": Intrinsically Fluorescent Polyureas toward Conformation-Assisted Metamorphosis, Discoloration and Intracellular Drug Delivery.

Response: We appreciate the helpful suggestion. Now the manuscript title has been replaced by "*Intrinsically Fluorescent Polyureas toward Conformation-Assisted Metamorphosis, Discoloration and Intracellular Drug Delivery*" according to the comment. Moreover, some similar statements such as "protein-inspired PUs" has also been changed to "*intrinsically fluorescent PUs*" in the revised manuscript (Page 3, Line 10)

2) Comment: Bibliography: The terms referring to peptides or amino acids are still not appropriate and should be replaced (pseudo-poly(amino acid)s, etc.). The term peptidomimetic is correct but I accept that it may be too broad. The reader should not be misled here. The article is not really dealing with amino acids (C monomer unit) and the polymer is PU. If the term peptidomimetic is not used, I suggest using the term PU-

based pseudo peptide to simplify/clarify the concept (for instance, p 3, lines 7-9). Finally, the references given in during the first revision should all be taken into account. Response: Thanks for the critical comment and thoughtful suggestions. We agree with the reviewer that the term peptidomimetic is more appropriate than pseudo-poly(amino acid). As suggested by the reviewer, the terms referring to pseudo-poly(amino acid)s or pseudo-PAAAs have been replaced with “*peptidomimetics*” (Page 2, Line 12; Page 3, Line 8), “*peptidomimetic polymers*” (Page 1, Line 8; Page 2, Line 10), or “*PU-based peptidomimetics*” (Page 19, Line 5; Page 23, Line 6) in the revised manuscript.

In addition, we have included all the recommended references (*Mol. Syst. Des. Eng.* 2018, 3, 364-375; *J. Am. Chem. Soc.* 2021, 143, 3697; *J. Mat. Sci.*, 2014, 49, 7339; *J. Am. Chem. Soc.* 2005, 127, 2156-2164; *Angew. Chem. Int. Ed.* 2010, 49, 1067-1070; *J. Am. Chem. Soc.* 2013, 135, 4884-4892; *J. Am. Chem. Soc.* 2019, 141, 14530-14533; *Proc. Natl. Acad. Sci. USA* 2016, 113, 3954-3959) in the revised manuscript (refs 13, 14, 20, 21, 22, 23, 51 and 56).

3) Comment: Figure 1c. Why not using intrinsic fluorescence instead? This would be more convincing given the results developed by the authors.

Response: According to the helpful suggestion, we have carried out additional CLSM imaging to capture the morphology of PU assemblies using the intrinsic fluorescence. Considering the logical order of the discussions (fluorescence after self-assembly), the new results have been provided in Supplementary Fig. 28 and discussed in the revised manuscript: “*The intrinsic fluorescence enabled direct observation of PU assemblies using CLSM. As seen in Supplementary Fig. 28, the fluorescent images present well-dispersed spherical and tube-like particles with hollow structures, which are consistent with TEM results (Fig. 1c).*” (Page 15, Lines 16-19).

4) Comment: The quality of TEM images is still poor and the presence of nanotubes is still not completely verifiable. The authors should surely nuance this interpretation by indicating that it is a hypothesis that remains to be verified (SAXS etc.). Nanofiber

formation is certainly the correct interpretation and the DLS measurement issues have not been properly accounted for.

Response: We agree with the reviewer that it is difficult to discriminate between nanofibers and nanotubes according to the DLS curves and TEM images. To address this issue, we have conducted cryo-SEM observation. The results show “*a number of open-ended tube-like particles with large aspect ratios (Fig. 2i and Supplementary Fig. 11), suggesting a possible hollow interior structure*” (Page 7, Lines 16-18).

Further, fluorescence probe technique has also been established as a powerful tool to study the assembled morphology (*Proc. Natl. Acad. Sci. USA*, 2005, 102, 2922-2927; *Angew. Chem. Int. Ed.* 2021, 60, 2 – 10; *Macromolecules*, 2020, 53, 5992-6001; *J. Am. Chem. Soc.* 2018, 140, 6604–6610; *Biomacromolecules* 2016, 17, 1026–1039; *J. Phys. Chem. B* 2007, 111, 14244–14249; *Chem. Rev.*, 2014, 114, 8883–8942; *ACS Nano* 2010, 4, 6805–6817). This method can differentiate nanotubes from nanofibers, because only assemblies with hollow structures (such as vesicles and tubes) can accommodate hydrophilic guest (*Biomacromolecules* 2016, 17, 1026–1039; *Adv. Funct. Mater.*, 2016, 26, 7652-7661; *Chem. Commun.*, 2015, 51, 3762-3765; *Chem. Rev.*, 2020, 120, 2347–2407; *J. Polym. Sci., Part A: Polym. Chem.*, 2008, 46, 2601-2611; *J. Controlled Release*, 2020, 326: 276-296). To this end, the PU assemblies were encapsulated with both rhodamine 6G (R6G) and doxorubicin hydrochloride (DOX HCl) as hydrophilic probes. As seen in Fig. 1d, “*with the incorporation of R6G in the PU dispersions, the fluorescence intensity of the dye was much lower than that of free R6G dissolved in water (Fig. 1d), demonstrating a self-quenching effect resulted from the high local concentration of dyes within the vesicular or tubular interior*” (Page 8, Lines 4-7) (*J. Am. Chem. Soc.* 2016, 138, 7508– 7511; *ACS Mater. Lett.* 2020, 2, 602-609; *Macromolecules*, 2020, 53, 5992-6001; *Angew. Chem. Int. Ed.* 2021, 60, 2 – 10; *J. Am. Chem. Soc.* 2018, 140, 6604–6610; *Biomaterials*, 2017, 116, 82-94). Moreover, “*the DOX fluorescence was detected in both emission spectra (Supplementary Fig. 14) and confocal laser scanning microscopy (CLSM) images with typical spherical and tube-like architectures (Fig. 1c), which agrees well with TEM observation. These results*

reveal that the polymeric assemblies are capable of providing hydrophilic pockets for loading water soluble agents” (Page 8, Lines 8-12) (*Soft Matter*, 2011,7, 662-669; *Biomacromolecules* 2014, 15, 3072–3082; *J. Controlled Release*, 2010, 142, 40-46; *Adv. Mater.*, 2011, 23, 2796-2801).

To further verify the morphology of PU assemblies, we have carried out small-angle X-ray scattering (SAXS) analyses on the PU solutions as suggested by the reviewer. The results were presented in Supplementary Fig. 13 and discussed in the revised manuscript: “Solution small-angle X-ray scattering (SAXS) provided more information on the morphology of PU assemblies. The scattering pattern of P₂ presents a gradient of approximately -2 at low q and corresponds to a classic model of vesicle (Supplementary Fig. 13a),⁹ while the SAXS data of P₄ shows regular oscillations on the decay, which could be reasonably fitted to a hollow cylinder model (Supplementary Fig. 13b).¹⁴” (Page 7, Lines 18-22; Page 8, Lines 1-2).

Furthermore, we have also performed additional CLSM experiment using the intrinsic fluorescence of PUs. We can clearly observe the spherical and tube-like structures of PU assemblies with hollow interior in spite of the inevitable aggregation of nanoparticles during experiment and the relatively low resolution of CLSM images. The new results were provided in Supplementary Fig. 28 and discussed in the revised manuscript: “The intrinsic fluorescence enabled direct observation of PU assemblies using CLSM. As seen in Supplementary Fig. 28, the fluorescent images present well-dispersed spherical and tube-like particles with hollow structures, which are consistent with TEM results (Fig. 1c)” (Page 15, Lines 16-19).

In addition, to make the conclusion more rigorous, we have replaced some statements “nanotubes” with “non-spherical”, “tube-like” or “nanotube-like” structures (Page 3, Line 13; Page 7, Line 14; Page 7, Line 17; Page 8, Line 1; Page 8, Line 10; Page 13, Line 10; Page 15, Line 18), and added some description in the revised manuscript according to the reviewer’s suggestion: “The formation of nanotube-like structures warrants further investigation” (Page 8, Lines 1-2). As for DLS measurement, it should be noted that our repeat experiments showed typical bimodal

size distributions for P₄ assemblies (Fig. 1b), which are similar to those reported for other nanotubes and nanofibers (*Nanoscale*, 2013, 5, 8577–8585; *Macromol. Chem. Phys.* 2015, 216, 439–449; *ChemistrySelect* 2020, 5, 12570–12581; *Angew.Chem. Int.Ed.* 2015, 54,9376–9380; *Nano Lett.* 2008, 8, 4221–4228; *Adv. Mater.* 2013, 25, 1170–1172). However, the size distribution determined by DLS cannot precisely reflect the real diameters and lengths of nanotube-like structures because DLS method handles anisotropic rod-like particles as spherical particles within a hydrodynamic approximation (*Macromol. Rapid Commun.* 2011, 32, 1518-1525; *J. Nanosci. Nanotechnol.* 2005, 5, 1045-1049). To address this issue, we estimated the size (diameter and length) distributions of P₄ assemblies according to TEM images. The results were provided in Supplementary Fig. 11 and analyzed in the revised manuscript: “while P₄ formed non-spherical structures with diameters and lengths in the range of 10-30 and 93-132 nm, respectively (Fig. 1c, Supplementary Fig. 10 and 11)” (Page 7, Lines 14-15).

Overall, to address the comment, we have provided new experimental results and analyses in Supplementary Fig. 11, 13 and 28, and added additional discussion and explanation in the revised manuscript (Page 7, Lines 14-22; Page 8, Lines 1-2; Page 8, Lines 4-7; Page 15, Lines 16-19).

5) Comment: Esi. Figure 20: The units are missing in the analysis presented (molar ellipticity). As it is, it is difficult to conclude on the basis of the data unless the non-racemized polymer is overlaid to compare.

Response: Thanks for pointing out this issue. Now the missing units of molar ellipticity have been added in the y-axis in both Fig. 2a in the manuscript and Supplementary Fig. 22 in ESI. Moreover, as suggested by the reviewer, we have also overlaid the CD curves of DL-PU_s with those of non-racemized PU_s for comparison (Supplementary Fig. 22).

6) Comment: Esi. Please provide the integrations on the NMR analyses (fig. 8 for instance but also for the others).

Response: According to the reviewer's suggestion, we have provided the integrations on the NMR analyses in Supplementary Table 1.

To Reviewer #2

1. Comment: The revision is ready for publication.

Response: We are grateful to the reviewer's support for our work.

Overall, we have carried out additional experiments, responded in detail to each point of the reviewer, and revised the manuscript according to the comments. Moreover, we have carefully checked the manuscript and supporting document for grammar problems and other mistakes, and revised them appropriately. **All changes made are in red color as shown** in the documents **“Revised manuscript” and “Highlighted revised supporting information”**. We hope, with these modifications and improvements, the new manuscript would be suitable for publication in *Nature communications*.

The change of manuscript and supporting information **were also listed as follows** (unless otherwise noted, all locations shown below are directed to the original manuscript):

1. Changes in Manuscript

- (1) Page 1, Line 1, “Bioinspired” was changed to “Intrinsically”.
- (2) Page 1, Line 8, “Pseudo-poly(amino acid)s” was changed to “Peptidomimetic polymers”.
- (3) Page 2, Line 10, “family of pseudo-PAAAs have also been developed by incorporating α -amino acids into the polymeric backbones via ester, carbonate, urethane, imide and other non-amide linkages,¹¹⁻¹³” was changed to “variety of synthetic peptidomimetic polymers (poly- β -peptide, polypeptoids, etc.) have also been developed via amino acid extension, polypeptide side chain substitution or

- backbone modification.¹¹⁻¹⁵”.
- (4) Page 2, Line 12, “which” was changed to “The peptidomimetics”.
 - (5) Page 2, Line 12, “facile synthesis,” was deleted.
 - (6) Page 2, Line 13, “enhanced solubility,” was added.
 - (7) Page 2, Line 14, “the presence of” was changed to “the incorporation of”.
 - (8) Page 2, Line 14, “limits the” was changed to “may compromise their.
 - (9) Page 2, Line 15, “of pseudo-PAAAs” was deleted.
 - (10) Page 3, Line 8, “pseudo-PAAAs” was changed to “peptidomimetics”.
 - (11) Page 3, Lines 8, “structures” was changed to “conformations”.
 - (12) Page 3, Line 10, “protein-inspired” was changed to “a family of intrinsically fluorescent”.
 - (13) Page 3, Line 13, “nanotubes” was changed to “nanotube-like structures”.
 - (14) Page 4, Line 10, “(Supplementary Table 1)” was added.
 - (15) Page 6, Line 5, “Supplementary Fig.7” was changed to “Supplementary Fig.13”.
 - (16) Page 7, Line 14, “vesicular structures” was changed to “vesicles”.
 - (17) Page 7, Line 14, “nanotubes (Fig. 1c, Supplementary Fig. 10).” was changed to “non-spherical structures with diameters and lengths in the range of 10-30 and 93-132 nm, respectively (Fig. 1c, Supplementary Fig. 10 and 11).”.
 - (18) Page 7, Line 16, “tubular structures” was changed to “tube-like particles”.
 - (19) Page 7, Line 16, “Supplementary Fig. 11” was changed to “Supplementary Fig. 12”.
 - (20) Page 7, Line 17, “suggesting a possible hollow interior structure. Solution small-angle X-ray scattering (SAXS) provided more information on the morphology of PU assemblies. The scattering pattern of P2 presents a gradient of approximately -2 at low q and corresponds to a classic model of vesicle (Supplementary Fig. 13a),⁹ while the SAXS data of P4 shows regular oscillations on the decay, which could be reasonably fitted to a hollow cylinder model (Supplementary Fig. 13b).¹⁴ The formation of nanotube-like structures warrants further investigation.” was added.
 - (21) Page 7, Line 19, “assembled PU solutions, a self-quenching effect was evidenced

- by the decline of fluorescence intensity of the dye (Fig. 1d).” was changed to “PU dispersions, the fluorescence intensity of the dye was much lower than that of free R6G dissolved in water (Fig. 1d), demonstrating a self-quenching effect resulted from the high local concentration of dyes within the vesicular or tubular interior.¹⁰
- (22) Page 7, Line 22, “Supplementary Fig. 12” was changed to “Supplementary Fig. 14”.
- (23) Page 8, Line 1, “tubular” was changed to “tube-like”.
- (24) Page 8, Line 5, “Supplementary Fig.13” was changed to “Supplementary Fig.15”.
- (25) Page 8, Line 14, “Supplementary Fig.14” was changed to “Supplementary Fig.16”.
- (26) Page 8, Line 14, “Supplementary Fig.15” was changed to “Supplementary Fig.17”.
- (27) Page 9, Line 3, “(Supplementary Fig. 16).³¹” was changed to “Supplementary Fig. 18).³³”.
- (28) Page 9, Line 9, “Supplementary Fig. 17” was changed to “Supplementary Fig. 19”.
- (29) Page 11, Line 10, “Figs. 18-19” was changed to “Figs. 20-21”.
- (30) Page 11, Line 11, “Supplementary Fig. 20” was changed to “Supplementary Fig. 22”.
- (31) Page 12, Line 5, “(Supplementary Fig. 21 and Supplementary Table 1)” was changed to “(Supplementary Fig. 23 and Supplementary Table 2)”.
- (32) Page 12, Line 22, “Supplementary Fig. 22” was changed to “Supplementary Fig. 24”.
- (33) Page 13, Line 9 “nanotube-to-vesicle transition” was changed to “transition from nanotube-like structures to vesicles”.
- (34) Page 13, Line 10, “Supplementary Fig. 23” was changed to “Supplementary Fig. 25”.
- (35) Page 13, Line 12, “Supplementary Fig. 24” was changed to “Supplementary Fig. 26”.
- (36) Page 13, Line 15, “Supplementary Fig. 24” was changed to “Supplementary Fig. 26”.

- (37) Page 15, Line 15, “Supplementary Fig. 25” was changed to “Supplementary Fig. 27”.
- (38) Page 15, Line 15, “The intrinsic fluorescence enabled direct observation of PU assemblies using CLSM. As seen in Supplementary Fig. 28, the fluorescent images present well-dispersed spherical and tube-like particles with hollow structures, which are consistent with TEM results (Fig. 1c).” was added.
- (39) Page 15, Line 15, “with” was changed to “The PUs showed”.
- (40) Page 15, Line 16, “Supplementary Fig. 26” was changed to “Supplementary Fig. 29”.
- (41) Page 15, Line 18, “Supplementary Fig. 27” was changed to “Supplementary Fig. 30”.
- (42) Page 16, Line 13, “Supplementary Fig. 28” was changed to “Supplementary Fig. 31”.
- (43) Page 18, Line 4, “Supplementary Fig. 29” was changed to “Supplementary Fig. 32”.
- (44) Page 18, Line 7, “Supplementary Fig. 30” was changed to “Supplementary Fig. 33”.
- (45) Page 18, Line 14, “Supplementary Fig. 31” was changed to “Supplementary Fig. 34”.
- (46) Page 18, Line 19, “Supplementary Fig. 32” was changed to “Supplementary Fig. 35”.
- (47) Page 20, Line 2, “pseudo-PAA” was changed to “PU-based peptidomimetics”.
- (48) Page 20, Line 6, “Supplementary Fig. 33” was changed to “Supplementary Fig. 36”.
- (49) Page 20, Line 14, “Supplementary Fig. 34, Supplementary Table 2” was changed to “Supplementary Fig. 37, Supplementary Table 3”.
- (50) Page 20, Line 16, “Supplementary Fig. 35” was changed to “Supplementary Fig. 38”.
- (51) Page 20, Line 22, “Supplementary Fig. 36” was changed to “Supplementary Fig. 39”.
- (52) Page 21, Line 2, “Supplementary Fig. 37” was changed to “Supplementary Fig. 40”.
- (53) Page 21, Line 21, “novel urea-constructed pseudo-PAA” was changed to “class of novel amino acid-constructed and PU-based peptidomimetics”.

In addition, Fig 2a was modified, new references (refs 13, 14, 15) were added and

the reference list was updated.

2. Changes in Supporting Information

- (1) Page 1, Line 3, “Bioinspired” was changed to “Intrinsically”.
- (2) Page 4, Line 34, “Confocal laser scanning microscope (CLSM). P1, P2, P3, and P4 assemblies (1 mg mL⁻¹) were added to the surface of a glass slide and sealed with cover glass, and then maintained at 4 °C for 12 h. The samples were imaged on a confocal laser scanning microscope (CLSM, Nikon A1RMP, Japan).” was added.
- (3) Page 4, Line 35, “To verify the morphology of self-assembled PUs,” was deleted.
- (4) Page 4, Line 35, “a” was changed to “A”.
- (5) Page 4, Line 39, “confocal laser scanning microscope (CLSM, Nikon A1RMP, Japan).” was changed to “CLSM.”.
- (6) Page 5, Line 5, “Small Angle X-ray Scattering (SAXS). SAXS measurements were conducted on a Xeuss 2.0 system (Xenocs SA, Grenoble, France) with a microfocused Cu K α source and a Rayonix MX225-HE CCD X-ray detector. The beamline was operated at 15 keV corresponding to a wavelength of 0.83 Å. For measurement, 100 μ L of PU dispersion in water was transferred into a quartz glass capillary with a diameter of 1.5 mm. The sample-to-detector distance was set as 3489.2 mm to collect data from $q = 0.0042 \sim 0.114 \text{ \AA}^{-1}$ range. The length of scattering vector q was defined as $q = 4 \sin \theta / \lambda$, where θ is half of the angle between incident and scattered X-rays, λ is the wavelength of the X-ray. Scattering from water was recorded in the same way for background subtraction. The single spectra were averaged and subtracted for background using the FOXTROT software. The SAXS curve was fitted with a SasView software using a hollow cylinder model and vesicle model.¹¹⁻¹³” was added.
- (7) Page 19, Line 3, “Supplementary Fig. 11 and its caption.” was added.
- (8) Page 20, Line 2, “Supplementary Fig. 13 and its caption.” was added.
- (9) Page 28, Line 2, “Supplementary Fig. 20 CD spectra of DL-PU assemblies: (a)

DL-P1; (b) DL-P4.” was changed to “Supplementary Fig. 22 CD spectra of PU and DL-PU assemblies.”.

(10) Page 34, Line 2, “Supplementary Fig. 28 and its caption.” was added.

(11) Page 46, Line 2, “Supplementary Table 1 and its caption.” was added.

In addition, new figures (Supplementary Figs 11, 13 and 28) were added; Supplementary Fig 22 was modified; The numbers of Supplementary Figs were updated. Moreover, new supporting references were added and the reference list was updated.

REVIEWERS' COMMENTS

Reviewer #1 (Remarks to the Author):

This revised manuscript basically addressed all my previous concerns. All the questions have been brightly addressed in this revision. I then recommend the manuscript for publication in Nature Communications and I greatly congrats all the authors for their contribution to the field and for their hard work.